# AN ENSEMBLE FRAMEWORK FOR UNBIASED LANGUAGE MODEL WATERMARKING

**Yihan Wu**[*], **Ruibo Chen**[*], **Georgios Milis**, **Heng Huang**[†]
Department of Computer Science
University of Maryland, College Park
{ywu42, rbchen, milis, heng}@umd.edu

## ABSTRACT

As large language models (LLMs) become increasingly capable and widely deployed, verifying the provenance of machine-generated content is critical to ensuring trust, safety, and accountability. Watermarking techniques have emerged as a promising solution by embedding imperceptible statistical signals into the generation process. Among them, unbiased watermarking is particularly attractive due to its theoretical guarantee of preserving the language model's output distribution, thereby avoiding degradation in fluency or detectability through distributional shifts. However, existing unbiased watermarking schemes often suffer from weak detection power and limited robustness, especially under short text lengths or distributional perturbations. In this work, we propose ENS, a novel ensemble framework that enhances the detectability and robustness of logits-based unbiased watermarks while strictly preserving their unbiasedness. ENS sequentially composes multiple independent watermark instances, each governed by a distinct key, to amplify the watermark signal. We theoretically prove that the ensemble construction remains unbiased in expectation and demonstrate how it improves the signal-to-noise ratio for statistical detectors. Empirical evaluations on multiple LLM families show that ENS substantially reduces the number of tokens needed for reliable detection and increases resistance to smoothing and paraphrasing attacks without compromising generation quality.

## 1 INTRODUCTION

The rapid progress of LLMs has enabled them to generate human-level text at scale, raising urgent questions about provenance, accountability, and misuse of AI-generated content. A leading line of defence is *watermarking*, which embeds hidden statistical signals into the generation process so that downstream detectors can later verify authorship with high confidence (Aaronson, 2022; Kirchenbauer et al., 2023; Christ et al., 2023; Kuditipudi et al., 2023; Hu et al., 2023; Wu et al., 2023; Chen et al., 2024a;b; Liu & Bu, 2024; Chen et al., 2025; Mao et al., 2024; Dathathri et al., 2024). Ideally, a watermark should satisfy three properties: *(i) imperceptibility*: it should leave fluency and semantics intact; *(ii) detectability*: it should be reliably identified from a moderate amount of text; and *(iii) robustness*: it should withstand natural corruptions and adversarial removal attempts.

Among these methods, a particularly important subclass is *unbiased* (a.k.a. distortion-free) watermarking (Aaronson, 2022; Christ et al., 2023; Kuditipudi et al., 2023; Hu et al., 2023; Wu et al., 2023; Chen et al., 2025; Mao et al., 2024; Dathathri et al., 2024), which modifies the generation distribution in a way that preserves its expectation. In other words, the average output distribution of a watermarked model remains indistinguishable from the original model, ensuring that watermarking does not degrade text quality or introduce detectable artifacts. This property makes unbiased watermarking especially attractive for real-world deployment, where imperceptibility and content fidelity are critical.

---

[*]Equal contributions

[†]This work was partially supported by NSF IIS 2347592, 2348169, DBI 2405416, CCF 2348306, CNS 2347617, RISE 2536663.

However, unbiased watermarking faces key limitations in practice. Because the expected distribution is unchanged, the statistical signal available to the detector is inherently weak. This often requires longer token sequences for detection and leaves the watermark more vulnerable to attacks such as sampling smoothing, truncation, or paraphrasing. To address these challenges, we propose ENS, an *ensemble* framework that composes multiple unbiased watermark instances with independent keys to amplify the detection signal while provably preserving the LM distribution. Intuitively, each watermark layer makes a tiny, unbiased perturbation; stacking $n$ such layers aggregates the bias under the detector's statistic, boosting signal-to-noise ratio roughly by $\sqrt{n}$ while keeping generation quality intact.

Our main contributions are summarized as follows:

- We introduce ENS, a general ensemble framework for unbiased watermarking. Our method sequentially applies multiple independent unbiased reweighting functions, using distinct keys, to enhance the detection signal without altering the expected distribution.
- We prove that ENS remains unbiased in expectation under independent keys, and we analyze how the ensemble size affects the watermark's detectability and robustness.
- We empirically demonstrate that ENS significantly improves detection performance and robustness to common perturbations across multiple model families and watermarking baselines, while preserving generation quality.

## 2 RELATED WORK

Kirchenbauer et al. (2023) refined the statistical watermarking framework initially introduced by Aaronson (2022). They divided the language model tokens into red and green lists and favored the green list tokens by adjusting their logits with a fixed increment $\delta$. However, the proposed approach can inevitably decrease the generation quality of the text. To maintain the original output distribution in watermarked content, researchers have investigated novel approaches for token distribution modification. There are generally two types of unbiased watermark: sampling based unbiased watermark and logits based unbiased watermark. For sampling based unbiased watermark, we use the pseudo-random token sampler to sample the next token. For logits based unbiased watermark, we adapt pseudo-random adjuster to modify the LM token logits, then sampling the next token based on the modified LM logits.

**Sampling based unbiased watermarks.** Aaronson (2022) pioneered an unbiased watermarking method using Gumbel-max to pseudo-randomly sample the next token with prefix n-grams as watermark keys. Christ et al. (2023) used inverse sampling as the watermark sampler on a binary language model with watermark keys based on token positioning. ITS-edit and EXP-edit Kuditipudi et al. (2023) refined the inverse-sampling and Gumbel-max strategies with a predetermined watermark key list. Hu et al. (2023) used inverse-sampling and design a model-based LLR score for detection. STA-1 Mao et al. (2024) adapted rejected-sampling strategy to improve the quality of the watermarked text under the low-entropy scenarios. Dathathri et al. (2024) proposed SynthID, which used multi-layer tournament sampling to achieved better detectability.

**Logits based unbiased watermarks.** Hu et al. (2023) introduced the first logits based unbiased watermark, $\gamma$-reweight, for watermarking, though their detection method is not model-agnostic. DiPmark Wu et al. (2023) enhanced the $\gamma$-reweight technique and introduced a model-agnostic detector. Chen et al. (2025) presented MCmark, which significantly improved the detectability of the unbiased watermark.

## 3 PRELIMINARIES

**Notation.** We follow the notation conventions from prior work (Hu et al., 2023; Wu et al., 2023; Mao et al., 2024; Chen et al., 2025) to describe the language model generation process. Let $V$ denote the vocabulary set with cardinality $N = |V|$. Define $\mathcal{V}$ as the set of all possible token sequences of any length (including the empty sequence). Given a prompt, the LM generates tokens autoregressively. The probability of generating the next token $x_{t+1} \in V$, conditioned on the preceding sequence $x_1, \ldots, x_t$, is denoted by $P_M(x_{t+1} \mid \boldsymbol{x}_{1:t}) \in \mathcal{P}$, where $\mathcal{P}$ represents the space of token distributions.

**Watermark Generator.**    To embed watermarks, the service provider modifies the original LM distribution $P_M$ using either sampling-based or logits-based watermarking methods. The watermarking process involves reweighting $P_M(\cdot \mid \boldsymbol{x}_{1:t})$ to a watermarked distribution $P_{M,w}(\cdot \mid \boldsymbol{x}_{1:t}, k)$, where $k \in \mathcal{K}$ is a private key sampled from a known key space distribution $P_{\mathcal{K}}(k)$. In practice, this key is typically derived from a hash function $h$ applied to a *secret key* sk and a context identifier (e.g., an $n$-gram (Aaronson, 2022) or token position (Christ et al., 2023)).

**Logits-Based Reweighting.**    In logits-based watermarking, the watermark generator applies a reweighting strategy $F \in \mathcal{F} : \mathcal{P} \times \mathcal{K} \to \mathcal{P}$, which transforms the original LM distribution $P_M(\cdot \mid x_{1:t})$ into a watermarked distribution $F(P_M(\cdot \mid x_{1:t}), k)$. Following (Hu et al., 2023), a reweighting function $F$ is said to be *unbiased* if it preserves the original distribution in expectation over the key space. Specifically, for any $P_M \in \mathcal{P}$ and $x_{t+1} \in V$,

$$\mathbb{E}_{k \sim P_{\mathcal{K}}} \left[ F(P_M(x_{t+1} \mid \boldsymbol{x}_{1:t}), k) \right] = P_M(x_{t+1} \mid \boldsymbol{x}_{1:t}). \tag{1}$$

**Watermark Detector.**    At inference time, the *detector* receives: (i) the generated text sequence $\boldsymbol{x}_{1:T}$, (ii) the watermark keys (or a means to regenerate them), and (iii) the public description of $F$. Detection is framed as a binary hypothesis testing problem: $H_0$: The sequence was generated without a watermark; $H_1$: The sequence was generated with a watermark. The detector constructs a statistical score function based on $k$ and $F$, which exhibits different distributions under $H_0$ and $H_1$. Under $H_1$, the statistic is stochastically larger (or smaller) than under $H_0$, enabling detection with controlled type-I error. This allows for distinguishing between watermarked and non-watermarked content using standard hypothesis testing techniques.

## 4 METHODOLOGY

In this section, we introduce an ensemble framework for logits-based unbiased watermarks, aiming to improve both detection reliability and robustness without sacrificing unbiasedness. We propose an *ensemble* construction that composes any base logits-based unbiased watermarking rule $F$ with multiple independently drawn watermark keys. Intuitively, each application of $F$ injects an (unbiased) weak statistical signal aligned with its key; composing $n$ such transforms stacks $n$ weak signals. Because unbiasedness is defined in expectation over the key distribution, sequential composition preserves marginal unbiasedness to the underlying LM distribution when keys are independent. At detection time, we exploit the availability of all $n$ watermark keys to aggregate evidence across them, yielding improved statistical power at the same (or controlled) false positive rate (FPR). This section formalizes the construction, proves unbiasedness, develops score aggregation and significance testing procedures, and discusses practical design choices (key scheduling, context partitioning, computational cost, and robustness considerations).

**Definition 4.1** (Watermark ensemble). *Given an original LM distribution $P_M(\cdot|\boldsymbol{x}_{1:t}) \in \mathcal{P}$, a logits-based reweight strategy $F \in \mathcal{F}$, and $n$ watermark keys $\boldsymbol{k}_{1:n}$, we define the $n$-fold ensemble transform* ENS $: \mathbb{N} \times \mathcal{P} \times \mathcal{F} \times \mathcal{K}^n \to \mathcal{P}$ *recursively ,*

$$\text{ENS}(n, F, P_M(\cdot|\boldsymbol{x}_{1:t}), \boldsymbol{k}_{1:n}) = \begin{cases} F(\text{ENS}(n-1, F, P_M(\cdot|\boldsymbol{x}_{1:t}), \boldsymbol{k}_{1:n-1}), k_n), & n > 1; \\ F(P_M(\cdot|\boldsymbol{x}_{1:t}), k_1), & n = 1. \end{cases} \tag{2}$$

*For brevity we suppress the conditioning context $\boldsymbol{x}_{1:t}$ when unambiguous and write* $\text{ENS}_n(P_M)$.

The ensemble mechanism can be viewed as a multi-layer reweighting pipeline, where each layer applies a subtle perturbation governed by its corresponding key. The resulting distribution remains close to $P_M$ in total variation, but the watermark signal accumulates across layers, enhancing the detectability of the watermark. The detailed watermarking algorithm can be found in Algs. 1 and 2.

**Sequential vs. parallel views.**    The recursive definition is *sequential*: each new key reweights the distribution output by the previous step. An equivalent *parallel* interpretation, which is useful for certain implementations, treats the final logits as a sum of per-key adjustments applied to the original logits before reweighting. Sequential and parallel forms are algebraically equivalent for common families of $F$ (e.g., additive logit shifts, multiplicative temperature scaling within greenlists).

### 4.1 UNBIASEDNESS OF THE ENSEMBLE

A crucial property of unbiased watermarking is the preservation of the original distribution's expectation over random keys. We show that this property holds under our ensemble construction:

**Theorem 4.2** (Unbiasedness). *If $F$ is an unbiased logits-based reweight strategy, and the watermark keys $\boldsymbol{k}_{1:n}$ are i.i.d. from $P_{\mathcal{K}}$, then the $n$-ensemble of $F$ is also an unbiased logits-based reweight strategy, i.e.,*

$$\mathbb{E}_{\boldsymbol{k}_{1:n} \sim P_{\mathcal{K}}^n}[\text{ENS}_n(P_M(\cdot|\boldsymbol{x}_{1:t}))] = P_M(\cdot \mid \boldsymbol{x}_{1:t}) \tag{3}$$

*holds for arbitrary $P_M(\cdot|\boldsymbol{x}_{1:t}) \in \mathcal{P}$,*

*Proof sketch.* We argue by induction on $n$. The base case ($n=1$) reduces to Eq. 1. Suppose Eq. 3 holds for $n-1$. For $n$, condition on the first $n-1$ keys and apply the law of iterated expectation:

$$\mathbb{E}_{\boldsymbol{k}_{1:n}}[\text{ENS}_n(P_M(\cdot|\boldsymbol{x}_{1:t}))(v)] = \mathbb{E}_{\boldsymbol{k}_{1:n-1}}\left[\mathbb{E}_{k_n}[F(\text{ENS}_{n-1}(P_M(\cdot|\boldsymbol{x}_{1:t})) \mid k_n)(v)]\right] \tag{4}$$

$$\overset{(a)}{=} \mathbb{E}_{\boldsymbol{k}_{1:n-1}}[\text{ENS}_{n-1}(P_M(\cdot|\boldsymbol{x}_{1:t}))(v)] \overset{(b)}{=} P_M(v|\boldsymbol{x}_{1:t}), \tag{5}$$

where $v$ is an arbitrary token in $V$, (a) applies Eq. 1 with $P = \text{ENS}_{n-1}(P)$ (valid because $F$ is unbiased *for any input distribution*), and (b) uses the induction hypothesis. $\square$

**Remarks.** (i) Independence is sufficient but not necessary: certain correlated key schedules also preserve unbiasedness if the marginal over the final key $k_n$ is uniform and $F$ obeys a linearity property. (ii) If $F$'s unbiasedness holds only approximately (e.g., due to numerical truncation or top-$K$ filtering), ensemble bias accumulates at most linearly in $n$.

**Watermark key design.** The unbiasedness of $\text{ENS}_n$ requires that the watermark keys be independent during *a single generation step*. Recall that a watermark key $k$ is typically derived as $h(\text{sk}, \text{n-gram})$, where $h$ is a hash function ensuring independence across different $(\text{sk}, \text{n-gram})$ pairs. Following this principle, there are two ways to generate independent keys for ensemble watermarking: a) Use $n$ distinct hash functions $h_1, \ldots, h_n$, so that in each generation step, $h_1(\text{sk}, \text{n-gram}), \ldots, h_n(\text{sk}, \text{n-gram})$ are independent. b) Use $n$ distinct secret keys $\text{sk}_1, \ldots, \text{sk}_n$, so that $h(\text{sk}_1, \text{n-gram}), \ldots, h(\text{sk}_n, \text{n-gram})$ remain independent. In our implementation, we use b) to ensure the independence of the watermark keys during generation.

### 4.2 DETECTION EFFICIENCY

The detection of the ensemble watermark is straightforward. If the ensemble is built upon a logit-based strategy $F$, we can simply apply the detection algorithm of $F$ $n$ times using the secret keys $(\text{sk}_1, \ldots, \text{sk}_n)$, and then aggregate the resulting detection scores.

**Detection with Ensemble Watermarks.** Given a generated sequence $\boldsymbol{x}_{1:T}$, a detector score function $S$, and secret keys $(\text{sk}_1, ..., \text{sk}_n)$, the detector computes per-key scores $\{S(\boldsymbol{x}_{1:T}, \text{sk}_i)\}_{i=1}^n$ and aggregates them, e.g. $S_{\text{ENS}}(\boldsymbol{x}_{1:T}) = \sum_{i=1}^n S(\boldsymbol{x}_{1:T}, \text{sk}_i)$. Because the watermark bias adds *coherently* across independent keys, $S_{\text{ENS}}$ enjoys a higher signal-to-noise ratio, enabling stronger hypothesis testing between $H_0$ (unwatermarked) and $H_1$ (watermarked).

**Signal and Variance Behavior.** Unbiasedness ensures that, a downstream observer without keys cannot distinguish $\text{ENS}_n(P)$ from $P$. Detection instead leverages *conditional* shifts introduced by each key. Let $S(x_t, \text{sk}_i)$ denote the per-token score computed by the detector when evaluated with secret key $\text{sk}_i$. Under the alternative hypothesis $H_1$ (watermarked with the same keys), the statistic $S(x_t, \text{sk}_i)$ has a mean shifted by a positive amount $\mu_i > 0$ relative to its mean under the null hypothesis $H_0$ (no watermark). The ensemble detector aggregates the shift across keys by combining either (a) raw per-key log-likelihood ratios, (b) standardized $z$-scores, or (c) non-parametric ranks.

Because generation uses *all* $n$ keys, the expected per-token shift in $S(x_t, \text{sk}_i)$ generally increases with $n$, but its variance also grows due to interactions among keys. For many logit-based reweighting $F$ of interest (greenlist reweighting (Kirchenbauer et al., 2023); DiP-style permute-reweight (Wu et al., 2023)), the per-key signals add approximately linearly at small $n$, yielding a signal-to-noise ratio (SNR) that scales roughly as $\sqrt{n}$ for the aggregated statistic. We formalize one such regime below.

**Proposition 4.3** (Approximate SNR scaling). *Assume (i) conditional independence of per-key centered scores given the underlying token, (ii) common per-key variance $\sigma^2$, and (iii) common mean shift $\mu$ under $H_1$. Then the sum statistic $S_{\mathrm{ENS}}(\boldsymbol{x}_{1:T}) = \sum_{i=1}^{n} S(\boldsymbol{x}_{1:T}, \mathsf{sk}_i)$ has mean $n\mu$ and variance $n\sigma^2$ under $H_1$, yielding $\mathrm{SNR}(S_{\mathrm{ENS}}) = \mu\sqrt{n}/\sigma$. Under $H_0$, $T$ has mean $0$ and variance $n\sigma_0^2$. Hence, for fixed FPR calibrated under $H_0$, power increases with $n$.*

**Example (DiPmark detector).** We illustrate the improved detecting strength of ensemble watermarking using the DiPmark detector. In the DiPmark detector, the red–green lists for generating each token are reconstructed using the secret key and the corresponding n-gram. The detector then counts the total number of green tokens in the sequence and applies a statistical test to determine watermark presence.

For a given key $\mathsf{sk}_i$, let $V_G(\boldsymbol{x}_{1:T}; \mathsf{sk}_i)$ denote the number of tokens in the sequence $\boldsymbol{x}_{1:T}$ that fall into the *green* set induced by $\mathsf{sk}_i$. The DiPmark score is

$$S_{\mathrm{DiP}}(\boldsymbol{x}_{1:T}, \mathsf{sk}_i) \;=\; \frac{V_G(\boldsymbol{x}_{1:T}; \mathsf{sk}_i)}{T} - 0.5.$$

Under the $H_0$ (unwatermarked text), the green indicators are i.i.d. Bernoulli$(1/2)$, so by Hoeffding's inequality the (one-sided) $p$-value is bounded by $p_{\mathrm{single}} \;\leq\; \exp\!\big(-2T\, S_{\mathrm{DiP}}(\boldsymbol{x}_{1:T}, \mathsf{sk}_i)^2\big)$.

Ensembling $n$ DiPmark detectors: Given keys $\mathsf{sk}_1, \ldots, \mathsf{sk}_n$, define the ensemble score

$$S_{\mathrm{ENS}}(\boldsymbol{x}_{1:T}) \;=\; \sum_{i=1}^{n} S_{\mathrm{DiP}}(\boldsymbol{x}_{1:T}, \mathsf{sk}_i) \;=\; \frac{\sum_{i=1}^{n} V_G(\boldsymbol{x}_{1:T}; \mathsf{sk}_i)}{T} - \frac{n}{2}.$$

Equivalently, writing $Z_{t,i} \in \{0,1\}$ for the green indicator of token $t$ under key $\mathsf{sk}_i$, we have $S_{\mathrm{ENS}}(\boldsymbol{x}_{1:T}) \;=\; n\!\left(\frac{1}{nT} \sum_{i=1}^{n} \sum_{t=1}^{T} Z_{t,i} - \frac{1}{2}\right)$. Assuming independence across $t$ and $i$ under the null, Hoeffding's inequality applied to the average over $nT$ bounded variables yields

$$p_{\mathrm{ENS}} \;=\; \Pr\!\big(S_{\mathrm{ENS}} \geq s\big) \;\leq\; \exp\!\left(-\frac{2T}{n}\, s^2\right) \quad \Longrightarrow \quad p_{\mathrm{ENS}} \;\leq\; \exp\!\left(-\frac{2T}{n}\, S_{\mathrm{ENS}}(\boldsymbol{x}_{1:T})^2\right).$$

In the special case where all single-key scores are equal, $S_{\mathrm{DiP}}(\boldsymbol{x}_{1:T}, \mathsf{sk}_1) = \cdots = S_{\mathrm{DiP}}(\boldsymbol{x}_{1:T}, \mathsf{sk}_n) = s_0$, we have $S_{\mathrm{ENS}} = ns_0$ and hence

$$p_{\mathrm{ENS}} \;\leq\; \exp\!\big(-2Tn\, s_0^2\big) \;=\; \Big[\exp\!\big(-2T\, s_0^2\big)\Big]^n \;\approx\; \big(p_{\mathrm{single}}\big)^n.$$

Thus, ensembling $n$ independent DiPmark detectors improves the exponent linearly in $n$, i.e., the $p$-value decays exponentially with $n$. In practice, the ensemble $p$-value need not decay exponentially in $n$, because the per-key DiPmark detectors can become less informative when used jointly (e.g., due to the attenuation of individual scores). Nevertheless, ensembling typically improves overall detection power (efficiency) and yields a smaller $p$-value than any single detector.

**Dealing with Key Dependence.** In practice keys may not be statistically independent across tokens because the keying function $h(\mathsf{sk}, c)$ often reuses a fixed secret $\mathsf{sk}$ with context $c$ derived from overlapping $n$-grams or positions. Overlapping contexts induce correlations in the green/boosted sets across ensemble members, which in turn correlate the per-key scores. To address repeated contexts $c$, we follow (Hu et al., 2023) and maintain a history of previously seen contexts during watermark generation; if the current $c$ already appears in the history, we bypass watermarking and sample from the original (unwatermarked) distribution.

### 4.3 EFFECT OF THE ENSEMBLE SIZE $n$ ON DETECTABILITY.

Increasing the ensemble size $n$ can strengthen detection by aggregating signal across keys, yielding (under standard independence assumptions) an exponentially decaying $p$-value in $n$. Denoted by $\gamma$ the portion of tokens promoted by the reweight algorithm, and $\epsilon$ the strength of promotion added on the logit. For logit–reweighting schemes that promote a per-key *green* subset of size $\gamma|V|$, the intersection of promoted sets shrinks as $\gamma^n|V|$, reducing the chance that any promoted token lies in the model's high-probability region, which makes the per-key effect attenuates and detectability saturates or even degrades.

Table 1: Detectability comparison of different watermarking methods under 250- and 500-token settings. We report True Positive Rates (TPR) at fixed False Positive Rates (FPR) of 0.1%, 0.01%, and 0.001%, along with the median $p$-values (lower is better).

| Watermarking Methods | 250 tokens | | | | 500 tokens | | | |
|---|---|---|---|---|---|---|---|---|
| | TPR@FPR | | | Median p-value$\downarrow$ | TPR@FPR | | | Median p-value$\downarrow$ |
| | 0.1%$\uparrow$ | 0.01%$\uparrow$ | 0.001%$\uparrow$ | | 0.1%$\uparrow$ | 0.01%$\uparrow$ | 0.001%$\uparrow$ | |
| DiPmark($\alpha$=0.3) | 41.48% | 32.22% | 23.28% | 4.48e-3 | 71.84% | 61.68% | 50.50% | 8.60e-6 |
| ENS-DiPmark($\alpha$=0.3, $n$=5) | 75.05% | 66.77% | 58.79% | 9.77e-7 | 96.30% | 91.51% | 86.62% | 3.28e-14 |
| ENS-DiPmark($\alpha$=0.3, $n$=10) | 74.49% | 63.21% | 55.18% | 2.49e-6 | 93.33% | 87.51% | 82.56% | 3.75e-14 |
| DiPmark($\alpha$=0.4) | 49.90% | 39.18% | 31.86% | 1.20e-3 | 80.48% | 70.94% | 62.17% | 1.27e-7 |
| ENS-DiPmark($\alpha$=0.4, $n$=5) | 74.74% | 67.15% | 58.52% | 5.36e-7 | 94.18% | 90.23% | 85.29% | 9.19e-15 |
| ENS-DiPmark($\alpha$=0.4, $n$=10) | 70.58% | 58.74% | 51.23% | 8.19e-6 | 91.75% | 85.99% | 79.59% | 1.19e-12 |
| $\gamma$-reweight | 52.00% | 42.02% | 33.99% | 7.47e-4 | 81.26% | 72.45% | 65.16% | 4.58e-8 |
| ENS-$\gamma$-reweight($n$=5) | 73.98% | 64.14% | 54.92% | 2.04e-6 | 94.12% | 88.58% | 83.59% | 4.81e-15 |
| ENS-$\gamma$-reweight($n$=10) | 65.81% | 55.54% | 48.56% | 1.75e-5 | 88.97% | 83.35% | 76.97% | 2.07e-11 |
| SynthID($m$=20) | 93.92% | 88.55% | 81.97% | 6.04e-12 | 99.68% | 98.60% | 97.64% | 2.83e-26 |
| SynthID($m$=30) | 93.83% | 88.36% | 83.50% | 1.91e-12 | 99.24% | 98.37% | 97.50% | 4.07e-28 |
| SynthID($m$=40) | 92.12% | 87.58% | 82.42% | 4.79e-12 | 99.24% | 97.84% | 96.11% | 7.70e-28 |
| MCMark($l$=20) | 94.36% | 90.37% | 86.07% | 4.18e-13 | 99.34% | 98.45% | 97.01% | 8.30e-26 |
| ENS-MCMark($l$=20, $n$=3) | **95.70%** | **91.71%** | **87.93%** | **1.43e-14** | **99.89%** | **99.57%** | **98.59%** | 2.58e-31 |
| ENS-MCMark($l$=20, $n$=5) | 95.15% | 91.44% | 85.88% | 4.27e-14 | 99.12% | 98.90% | 98.02% | **1.27e-35** |

Let $|V|$ be the vocabulary size. For secret key $\mathsf{sk}_i$, let $G_i \subset [|V|]$ be the green set with $|G_i| = \gamma|V|$ $(0 < \gamma < 1)$. Under an *intersection-at-generation* scheme, promotions apply to $G^{(\cap)} = \bigcap_{i=1}^{n} G_i$, so $\mathbb{E}\big[|G^{(\cap)}|\big] \approx \gamma^n|V|$. Let $p(\cdot)$ denote the pre-boost next-token distribution at a step, and define the *promoted mass* $M_n := \sum_{v \in G^{(\cap)}} p(v)$.

Since $\sum_v p(v) = 1$ and each token lies in $G^{(\cap)}$ with probability $\gamma^n$, we have $\mathbb{E}[M_n] = \gamma^n$. If a logit boost of size $\varepsilon \leq 1/\gamma$ (if $\varepsilon > 1/\gamma$, then the sum of the boosted probability will be greater than 1) is applied to each promoted token during each logit-based reweighting, the per-step shift in a DiPmark-style score satisfies $\mu(n) \approx \varepsilon^n \mathbb{E}[M_n] = (\varepsilon\gamma)^n$.

**Aggregation gain vs. sparsity loss.** Let $S_i$ be the per-key detector score and $S_{\mathrm{ENS}} = \sum_{i=1}^{n} S_i$. Under standard independence assumptions and boundedness of scores, a Hoeffding/Chernoff bound yields $p_{\mathrm{ENS}}(\boldsymbol{x}_{1:n}) \lesssim \exp\big(-CTn\mu(n)^2\big) = \exp\big(-CTn(\varepsilon\gamma)^{2n}\big)$, for some constant $C > 0$ and sequence length $T$. This expression makes the trade-off explicit:

$$\underbrace{n}_{\text{aggregation gain}} \quad \text{vs.} \quad \underbrace{(\varepsilon\gamma)^{2n}}_{\text{promotion sparsity}} \quad .$$

Define $g(n) := n\,(\varepsilon\gamma)^{2n}$. Then $g(n)$ increases only up to $n^\star \approx \frac{1}{2\log(1/\varepsilon\gamma)}$, and decreases thereafter. Consequently, $p_{\mathrm{ENS}}$ typically *decreases* with $n$ for $n \lesssim n^\star$ (improving detectability), but *stalls* and can effectively *worsen* for $n \gg n^\star$ as the promoted mass $\mathbb{E}[M_n] = \gamma^n$ becomes vanishingly small (in the extreme, $\gamma^n|V| \lesssim 1$ so no token is promoted at many steps).

**Design implication.** In practice, choose a *moderate* ensemble size $n$ so that $n \approx n^\star$. For instance, if $\gamma = 0.5, \varepsilon = 1.8$, then $n^\star \approx 1/(2\log(1/0.9)) \approx 4.75$, suggesting $n \in \{4,5\}$ is near optimal under strict intersection. Larger $n$ can be viable only if the ensemble design avoids strict intersection (e.g., by aggregating per-key logits or statistics) so that the per-key effect size $\mu$ does not collapse with $n$.

## 5 EXPERIMENTS

Our experiments comprise three main parts. First, we evaluate the detectability gains of our ensemble watermark by comparing it with other unbiased watermarking methods on a text generation task. Second, we assess robustness under random token modifications, DIPPER paraphrasing attacks (Krishna et al., 2023), GPT paraphrasing attacks and GPT back translation attacks. Finally, we verify

Table 2: Robustness comparison under paraphrasing attacks using GPT-4o-mini and DIPPER.

| Watermarking Methods | GPT-4o-mini paraphrase | | | | DIPPER | | | |
|---|---|---|---|---|---|---|---|---|
| | TPR@FPR | | | Median p-value↓ | TPR@FPR | | | Median p-value↓ |
| | 0.1%↑ | 0.01%↑ | 0.001%↑ | | 0.1%↑ | 0.01%↑ | 0.001%↑ | |
| ENS-DiPmark($\alpha$=0.3, $n$=5) | 8.41% | 5.14% | 2.80% | 3.15e-1 | 3.26% | 1.09% | 0.00% | 2.61e-1 |
| ENS-DiPmark($\alpha$=0.4, $n$=5) | 7.11% | 3.56% | 3.56% | 2.13e-1 | 4.38% | 1.03% | 0.77% | 3.55e-1 |
| ENS-$\gamma$-reweight($n$=5) | 6.73% | 3.59% | 3.14% | 2.78e-1 | 3.05% | 1.39% | 0.83% | 3.97e-1 |
| SynthID($m$=30) | 25.39% | 13.47% | 6.74% | 1.72e-2 | 21.58% | 11.05% | 7.11% | 2.06e-2 |
| ENS-MCMark($l$=20, $n$=3) | **40.00%** | **29.44%** | **20.56%** | **3.69e-3** | **40.35%** | **30.70%** | **22.22%** | **5.09e-3** |

Table 3: Robustness comparison under GPT-4o-mini back translation (English–French) attacks using GPT-4o-mini and 10% random token replacement.

| Watermarking Methods | Back translation (En–Fr) | | | | 10% Random token replacement | | | |
|---|---|---|---|---|---|---|---|---|
| | TPR@FPR | | | Median p-value↓ | TPR@FPR | | | Median p-value↓ |
| | 0.1%↑ | 0.01%↑ | 0.001%↑ | | 0.1%↑ | 0.01%↑ | 0.001%↑ | |
| ENS-DiPmark($\alpha$=0.3, $n$=5) | 38.74% | 26.31% | 18.74% | 4.61e-3 | 38.74% | 26.31% | 18.74% | 4.61e-3 |
| ENS-DiPmark($\alpha$=0.4, $n$=5) | 44.38% | 29.83% | 20.99% | 2.35e-3 | 44.38% | 29.83% | 20.99% | 2.35e-3 |
| ENS-$\gamma$-reweight($n$=5) | 41.71% | 28.86% | 21.42% | 3.53e-3 | 41.71% | 28.86% | 21.42% | 3.53e-3 |
| SynthID($m$=30) | 75.69% | 64.53% | 55.58% | 3.16e-6 | 75.69% | 64.53% | 55.58% | 3.16e-6 |
| ENS-MCMark($l$=20, $n$=3) | **84.17%** | **76.43%** | **68.53%** | **1.94e-8** | **84.17%** | **76.43%** | **68.53%** | **1.94e-8** |

the unbiasedness of our method by showing that its output quality on machine translation and text summarization tasks closely matches that of the unwatermarked baseline. All experiments are conducted on NVIDIA A6000 GPUs. All watermarking algorithms introduced negligible computation cost during LLM generation process. Detailed experimental settings are provided in Appendix D.

**Baselines.**  We evaluate our method against several baselines, including three logit-based unbiased watermarking algorithms: $\gamma$-reweight (Hu et al., 2023), DiPmark (Wu et al., 2023), and MCmark (Chen et al., 2025), as well as one sampling-based unbiased watermarking algorithm (Dathathri et al., 2024). While other sampling-based unbiased methods such as ITS-edit (Kuditipudi et al., 2023) and STA-1 (Mao et al., 2024) exist, prior work (Chen et al., 2025) has shown that they perform worse than MCmark, and therefore we omit them from our experiments.

**Models and Datasets.**  We employ Llama-3.2-3B-Instruct (Dubey et al., 2024), Mistral-7B-Instruct-v0.3 (Jiang et al., 2023), and Phi-3.5-mini-instruct (Abdin et al., 2024) for text generation tasks to assess the effectiveness of our proposed ENS. Following prior work (Kirchenbauer et al., 2023; Hu et al., 2023), we conduct experiments on a subset of the C4 dataset (Raffel et al., 2020). In addition, we include evaluations on three MMW datasets (Piet et al., 2023), Dolly CW (Conover et al., 2023), and two tasks from WaterBench (Tu et al., 2023).

For unbiasedness validation, we adopt the settings from Hu et al. (2023); Wu et al. (2023), using MBart (Liu, 2020) for machine translation and BART (Lewis, 2019) for text summarization. In the machine translation experiments, we use the WMT16 ro-en dataset (Bojar et al., 2016). For text summarization, we use the CNN/DailyMail dataset (See et al., 2017).

**Watermarking parameters.**  We evaluate the detectability of ENS on the text generation task with different language models. We generate 1,000 examples for each task. We use the prefix 2-gram together with a secret key as the watermark keys. We select $\alpha \in \{0.3, 0.4\}$ for DiPmark (Wu et al., 2023), tournament sampling layers $m \in \{20, 30, 40\}$ for SynthID (Dathathri et al., 2024), $l = 20$ for MCmark (Chen et al., 2025). For $\gamma$-reweight (Hu et al., 2023), we follow the settings in the original papers. For watermark ensemble we select $n \in \{1, 5, 10\}$ for DiPmark and $\gamma$ reweight and $n \in \{1, 3, 5\}$ for MCmark. We report true positive rate under x% theoretical guaranteed false positive rate (TPR@FPR) and the Median p-value.

## 5.1 DETECTABILITY

The results in Table 1 clearly demonstrate that our ensemble strategy consistently enhances the detectability of logit-based watermarking methods. For DiPmark and $\gamma$-reweight, applying the ensemble

Table 4: Unbiasedness comparison of different watermarking methods on text summarization and machine translation tasks.

| Watermarking Methods | Text Summarization | | | | Machine Translation | |
|---|---|---|---|---|---|---|
| | ROUGE-1 | ROUGE-2 | ROUGE-L | BERTScore | BLEU | BERTScore |
| No Watermark | 0.3768 | 0.1327 | 0.2379 | 0.3175 | 20.35 | 0.5576 |
| DiPmark($\alpha$=0.3) | 0.3767 | 0.1325 | 0.2384 | 0.3170 | 20.44 | 0.5583 |
| ENS-DiPmark($\alpha$=0.3, $n$=5) | 0.3760 | 0.1317 | 0.2375 | 0.3163 | 20.24 | 0.5555 |
| ENS-DiPmark($\alpha$=0.3, $n$=10) | 0.3768 | 0.1328 | 0.2383 | 0.3167 | 20.21 | 0.5588 |
| DiPmark($\alpha$=0.4) | 0.3768 | 0.1330 | 0.2385 | 0.3178 | 20.35 | 0.5559 |
| ENS-DiPmark($\alpha$=0.4, $n$=5) | 0.3768 | 0.1326 | 0.2380 | 0.3166 | 20.36 | 0.5585 |
| ENS-DiPmark($\alpha$=0.4, $n$=10) | 0.3756 | 0.1322 | 0.2379 | 0.3163 | 20.38 | 0.5590 |
| $\gamma$-reweight | 0.3767 | 0.1320 | 0.2376 | 0.3165 | 20.54 | 0.5588 |
| ENS-$\gamma$-reweight($n$=3) | 0.3769 | 0.1331 | 0.2385 | 0.3169 | 20.23 | 0.5577 |
| ENS-$\gamma$-reweight($n$=5) | 0.3759 | 0.1321 | 0.2378 | 0.3157 | 20.45 | 0.5579 |
| SynthID($m$=20) | 0.3761 | 0.1323 | 0.2380 | 0.3169 | 19.82 | 0.5559 |
| SynthID($m$=30) | 0.3776 | 0.1331 | 0.2387 | 0.3175 | 20.15 | 0.5582 |
| SynthID($m$=40) | 0.3774 | 0.1336 | 0.2382 | 0.3173 | 20.28 | 0.5566 |
| ENS-MCMark($l$=20, $n$=1) | 0.3769 | 0.1329 | 0.2386 | 0.3176 | 19.83 | 0.5543 |
| ENS-MCMark($l$=20, $n$=3) | 0.3767 | 0.1325 | 0.2380 | 0.3170 | 20.43 | 0.5589 |
| ENS-MCMark($l$=20, $n$=5) | 0.3769 | 0.1333 | 0.2388 | 0.3177 | 20.19 | 0.5631 |

scheme substantially boosts TPR across all false positive rate thresholds and reduces median $p$-values, confirming that aggregation over multiple keys strengthens statistical power. More importantly, when comparing against strong baselines such as SynthID and MCMark, our ensemble framework achieves state-of-the-art performance. In particular, ENS-MCMark reaches the highest TPRs and the lowest $p$-values under both 250- and 500-token settings, surpassing all competing methods and establishing our ensemble method as the most effective approach for watermark detectability

## 5.2 ROBUSTNESS

To comprehensively evaluate robustness, we conduct experiments under 4 challenging text corruption attacks: GPT-4o-mini paraphrasing, DIPPER paraphrasing, GPT-4o-mini English–French back translation, and 10% random token replacement. These transformations substantially alter surface forms while preserving semantics, providing a rigorous stress test for watermark detectability. As shown in Tables 2 and 3, all watermarking methods experience degraded performance under these attacks. Nevertheless, our ensemble framework consistently yields significant improvements, with ENS-MCMark achieving the highest TPR across all FPR thresholds and the lowest $p$-values in every attack scenario. These results highlight that, even under strong paraphrasing and token-level perturbations, our ensemble method maintains state-of-the-art detectability, clearly outperforming existing baselines.

## 5.3 UNBIASEDNESS

To assess the unbiasedness of watermarking, we evaluate generation quality across two representative tasks: text summarization and machine translation, using multiple standard metrics. For summarization, we report ROUGE-1/2/L and BERTScore, while for translation we adopt BLEU and BERTScore. See Appendix D for a detailed introduction of the metrics. As shown in Table 4, all watermarking methods, including our ensemble variants, achieve scores that are nearly identical to the no-watermark baseline. This indicates that, similar to other unbiased watermarking approaches, our ensemble framework does not degrade generation quality. The consistency across diverse metrics and tasks confirms that the improved detectability of our ensemble method comes without sacrificing semantic fidelity or fluency of the generated outputs.

## 5.4 ABLATION STUDY

In this section, we study the effect of generation length and ensemble size on detectability. All experiments are conducted on Llama-3.2-3B-Instruct using the C4 subset, and we report TPR@0.01% FPR together with the median $p$-value.

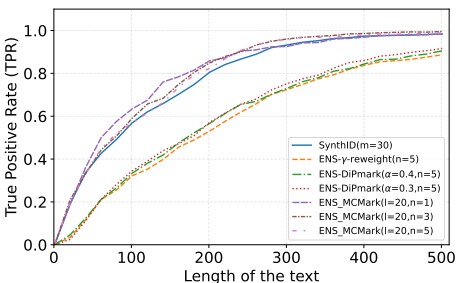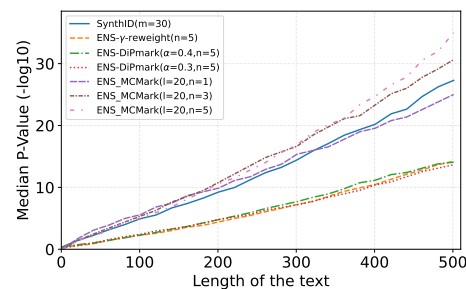

Figure 1: Effect of the generation length on the detectability. Left: TPR@0.01%FPR vs. generation length. Right: median $p$-value vs. generation length.

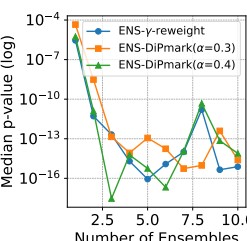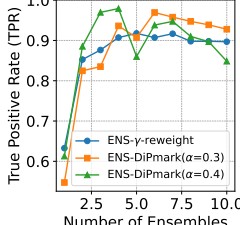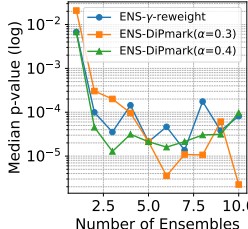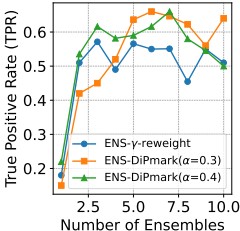

Figure 2: Effect of the number of ensembles on detectability. We compare $\gamma$-reweight and DiPmark ($\alpha = 0.3, 0.4$) under generation lengths 250 and 500. Left two plots: length=500; right two plots: length=250.

**Detectability vs. generation length.** As shown in Figure 1, increasing the generation length consistently improves detectability for all watermarking methods. Longer sequences provide more statistical evidence, thereby reducing the variance of detection scores and leading to both higher TPR and lower median $p$-values. Among the baselines, we observe that ENS-MCmark with $n = 3$ achieves the best overall detectability across lengths. Compared to the original MCmark baseline (*ENS-MCmark*($\ell = 20, n = 1$)), our ensemble framework significantly boosts detection power while maintaining robustness. These results highlight the importance of sequence length in watermark detection and further confirm the benefit of ensemble-based designs.

**Detectability vs. number of ensembles.** We further analyze the effect of ensemble size using DiPmark and $\gamma$-reweight with $n = 1, 2, \ldots, 10$ under generation lengths of 250 and 500 (Figure 2). Interestingly, detectability does not grow monotonically with $n$: we observe that detection power initially improves as ensembles aggregate complementary evidence, but then gradually declines when $n$ becomes large. This non-monotonic trend is consistent with our theoretical analysis in Sec. 4.3, where excessive ensemble averaging introduces redundancy and dilutes the effective signal. In particular, moderate ensemble sizes (e.g., $n = 3$–5) provide the best trade-off, achieving the lowest $p$-values and the highest TPR across both generation lengths.

## 6 CONCLUSION

In this work, we introduced ENS, a principled ensemble framework for unbiased watermarking that amplifies detection signals while rigorously preserving the underlying language model distribution. By composing multiple independent unbiased watermarks, ENS achieves a provable $\sqrt{n}$ gain in signal-to-noise ratio without sacrificing imperceptibility. Our theoretical analysis confirms that the unbiasedness property holds under independent keys, and our experiments demonstrate consistent improvements in detection accuracy and robustness to text modification attacks across diverse model families and baseline methods. These results suggest that ENS offers a practical and scalable path toward stronger, more reliable watermarking for real-world deployment, helping ensure provenance and accountability in the era of large-scale AI text generation.

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

## A  LLM Usage

We ONLY used ChatGPT-4o and ChatGPT-5 to refine the content.

## B  Watermarking Algorithms

---

**Algorithm 1** ENS generator.

---

1: **Input:** pretrained LM $P_M$, secret keys $\mathsf{sk}_1, \cdots, \mathsf{sk}_n$, prompt $\boldsymbol{x}_{-m:0}$, generate length $T \in \mathbb{N}$, n-gram window length $a$, logits reweight strategy $F$, hash function $h$, n-gram history $hist$.
2: **for** $t = 1, \ldots, T$ **do**
3:      Initialize $P_{M,w}^{(0)} = P_M$
4:      **if** $\boldsymbol{x}_{t-a,t-1} \in hist$ **then**
5:          Sampling from the original distribution $P_M(\cdot | \boldsymbol{x}_{-m:t-1})$.
6:      **else**
7:          Update $hist$ with $\boldsymbol{x}_{t-a,t-1}$.
8:          **for** $i = 1, \ldots, n$ **do**
9:              Generate watermark key $k_i = h(\mathsf{sk}_i, \boldsymbol{x}_{t-a,t-1})$.
10:              $P_{M,w}^{(i)}(\cdot | \boldsymbol{x}_{-m:t-1}) := F(P_{M,w}^{(i-1)}(\cdot | \boldsymbol{x}_{-m:t-1}) | k_i)$.
11:          Sample the next token $x_t$ from $P_{M,w}^{(n)}(\cdot | \boldsymbol{x}_{-m:t-1})$.
12: **return** $\boldsymbol{x}_{1:T}$.

---

**Algorithm 2** ENS detector.

---

1: **Input:** pretrained LM $P_M$, secret keys $\mathsf{sk}_1, \cdots, \mathsf{sk}_n$, generated tokens $\boldsymbol{x}_{1:T}$, threshold $\Phi_0$, score function $s$, logits reweight strategy $F$, hash function $h$.
2: Initialize $\Phi = 0$
3: **for** $t = 1, \ldots, T$ **do**
4:      **for** $i = 1, \ldots, n$ **do**
5:          Recover the watermark key $k_i = h(\mathsf{sk}_i, \boldsymbol{x}_{t-a,t-1})$.
6:          $\Phi = \Phi + s(x_t | F, k_i)$.
7: **if** $\Phi \geq \Phi_0$ **then**
8:      **return** $\boldsymbol{x}_{1:T}$ is watermarked.
9: **else**
10:      **return** $\boldsymbol{x}_{1:T}$ is not watermarked.

---

## C  Missing Proofs

**Theorem C.1** (Unbiasedness). *If $F$ is an unbiased logits-based reweight strategy, and the watermark keys $\boldsymbol{k}_{1:n}$ are i.i.d. from $P_{\mathcal{K}}$, then the $n$-ensemble of $F$ is also an unbiased logits-based reweight strategy, i.e.,*

$$\mathbb{E}_{\boldsymbol{k}_{1:n} \sim P_{\mathcal{K}}^n}[\text{ENS}(n, F, P_M(x_{t+1} | \boldsymbol{x}_{1:t}), \boldsymbol{k}_{1:n})] = P_M(x_{t+1} | \boldsymbol{x}_{1:t}) \quad (6)$$

*holds for arbitrary $P_M(\cdot | \boldsymbol{x}_{1:t}) \in \mathcal{P}$ and $x_{t+1} \in V$,*

*Proof.* We prove it by induction, when $n = 1$, since $F$ is an unbiased logit-based reweight strategy, we have

$$\mathbb{E}_{k \sim P_{\mathcal{K}}}[\text{ENS}(1, F, P_M(x_{t+1} | \boldsymbol{x}_{1:t}), k)] = \mathbb{E}_{k \sim P_{\mathcal{K}}}[F(P_M(x_{t+1} | \boldsymbol{x}_{1:t} | k))]$$
$$= P_M(x_{t+1} | \boldsymbol{x}_{1:t}). \quad (7)$$

When $m > 1$, assuming

$$\mathbb{E}_{\boldsymbol{k}_{1:n} \sim P_{\mathcal{K}}^n}[\text{ENS}(n, F, P_M(x_{t+1} | \boldsymbol{x}_{1:t}), \boldsymbol{k}_{1:n})] = P_M(x_{t+1} | \boldsymbol{x}_{1:t})$$

holds for $n = m - 1$, when $n = m$, since $k_m$ is independent of $\boldsymbol{k}_{1:m-1}$,

$$
\begin{aligned}
&\mathbb{E}_{\boldsymbol{k}_{1:m} \sim P_{\mathcal{K}}^{m}}[\text{ENS}(m, F, P_M(x_{t+1}|\boldsymbol{x}_{1:t}), \boldsymbol{k}_{1:m})] \\
=&\mathbb{E}_{k_m \sim P_{\mathcal{K}}, \boldsymbol{k}_{1:m-1} \sim P_{\mathcal{K}}^{m-1}}[F(\text{ENS}(m-1, F, P_M(\cdot|\boldsymbol{x}_{1:t}), \boldsymbol{k}_{1:m-1})|k_m)], \\
=&\mathbb{E}_{\boldsymbol{k}_{1:m-1} \sim P_{\mathcal{K}}^{m-1}}[\mathbb{E}_{k_m \sim P_{\mathcal{K}}}[F(\text{ENS}(m-1, F, P_M(\cdot|\boldsymbol{x}_{1:t}), \boldsymbol{k}_{1:m-1})|k_m)]]
\end{aligned}
\tag{8}
$$

Since $F$ is an unbiased logits-based reweight strategy, we have

$$
\mathbb{E}_{k_m \sim P_{\mathcal{K}}}[F(\text{ENS}(m-1, F, P_M(\cdot|\boldsymbol{x}_{1:t}), \boldsymbol{k}_{1:m-1})|k_m)] = \text{ENS}(m-1, F, P_M(\cdot|\boldsymbol{x}_{1:t}), \boldsymbol{k}_{1:m-1}).
$$

Together with Eq. 8,

$$
\begin{aligned}
&\mathbb{E}_{\boldsymbol{k}_{1:m} \sim P_{\mathcal{K}}^{m}}[\text{ENS}(m, F, P_M(x_{t+1}|\boldsymbol{x}_{1:t}), \boldsymbol{k}_{1:m})] \\
=&\mathbb{E}_{\boldsymbol{k}_{1:m-1} \sim P_{\mathcal{K}}^{m-1}}[\text{ENS}(m-1, F, P_M(\cdot|\boldsymbol{x}_{1:t}), \boldsymbol{k}_{1:m-1})], \\
\stackrel{(*)}{=}&P_M(x_{t+1} \mid \boldsymbol{x}_{1:t}).
\end{aligned}
\tag{9}
$$

(*) refers by induction assumption. Thus, the $n$-ensemble of $F$ is also an unbiased logits-based reweight strategy. □

## D EXPERIMENT SETUP

We assess the unbiasedness properties of various watermarking models across two seq2seq tasks: text summarization and machine translation. The experiments are implemented using the Huggingface library (Wolf et al., 2019), a widely adopted framework in the NLP community for model training and sharing. All evaluations are performed on 8 NVIDIA A6000 GPUs, each equipped with 48GB of memory.

**Machine Translation.** For this task, we use the WMT'14 English (En) to Romanian (Ro) dataset, which includes 1,999 test examples. The Multilingual BART (MBart) model (Liu, 2020) is adopted, along with its official tokenizer.

**Text Summarization.** For summarization, we utilize the CNN-DM test set (Hermann et al., 2015), consisting of 11,490 examples. We evaluate with the BART-large model (400M parameters) and the LLaMA-2 model with 7B parameters.

**Evaluation Metrics for Text Quality.** To quantify generation quality, we adopt the following metrics:

- **ROUGE Score.** Applied to summarization, ROUGE (Lin, 2004) measures n-gram overlap between generated summaries and reference texts, reflecting content preservation.

- **BLEU Score.** For translation, BLEU (Papineni et al., 2002) evaluates lexical similarity between system outputs and human references.

- **BERTScore.** BERTScore (Zhang et al., 2019) computes semantic similarity via token embedding cosine similarity. We report BERTScore-F1, -Precision, and -Recall for both summarization and translation.

- **Perplexity.** Derived from information theory, perplexity measures how well a probability model predicts text. Lower values indicate stronger predictive capacity. We use it to assess both summarization and text generation.

## E ADDITIONAL RESULTS

In this section, we provide all the experimental results on three LMs: Llama-3.2-3B-Instruct (Dubey et al., 2024), Mistral-7B-Instruct-v0.3 (Jiang et al., 2023), and Phi-3.5-mini-instruct (Abdin et al., 2024); and 7 datasets: C4 subset (Raffel et al., 2020), three MMW datasets (Piet et al., 2023), Dolly CW (Conover et al., 2023), and two datasets from WaterBench (Tu et al., 2023).

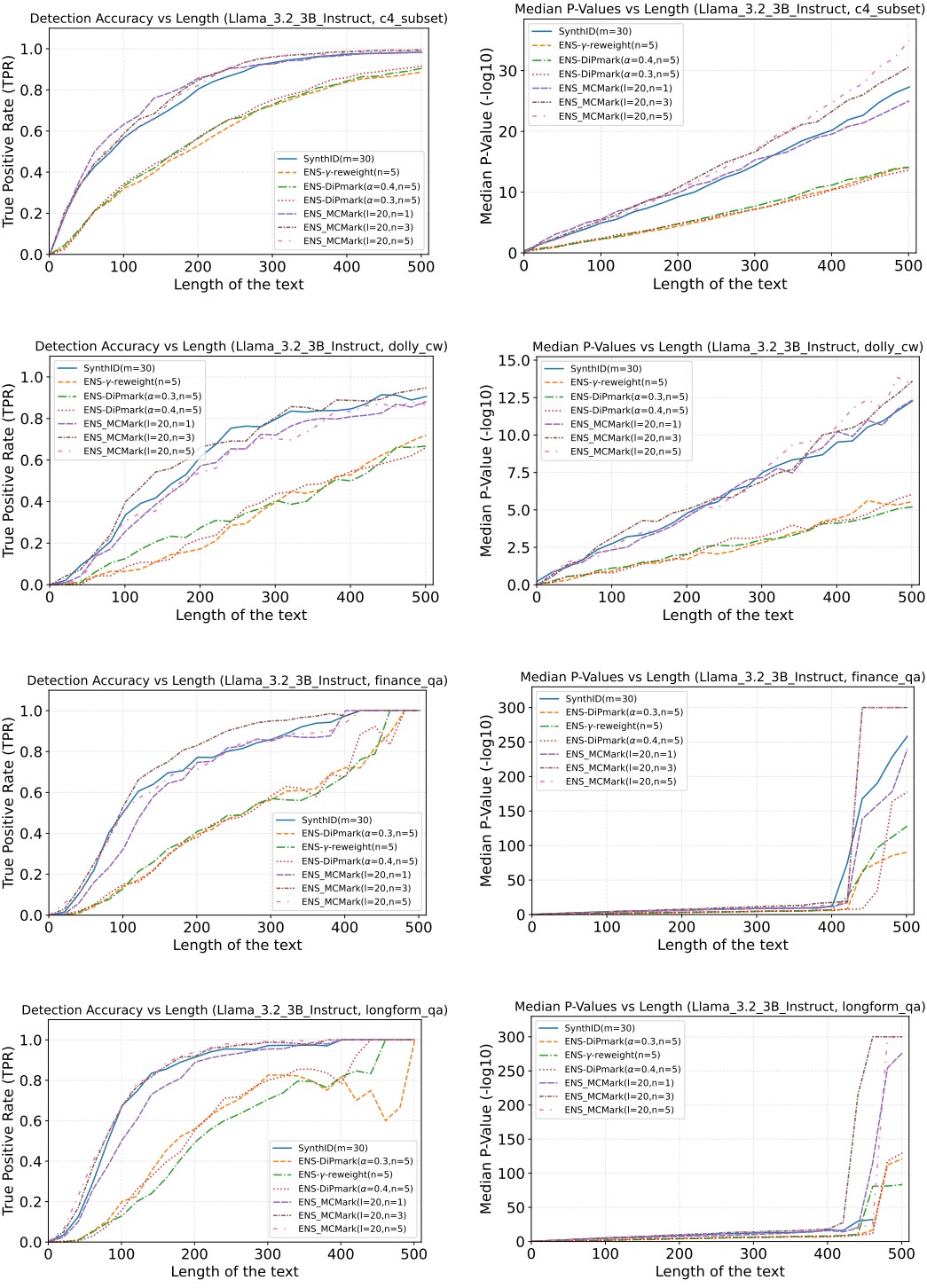

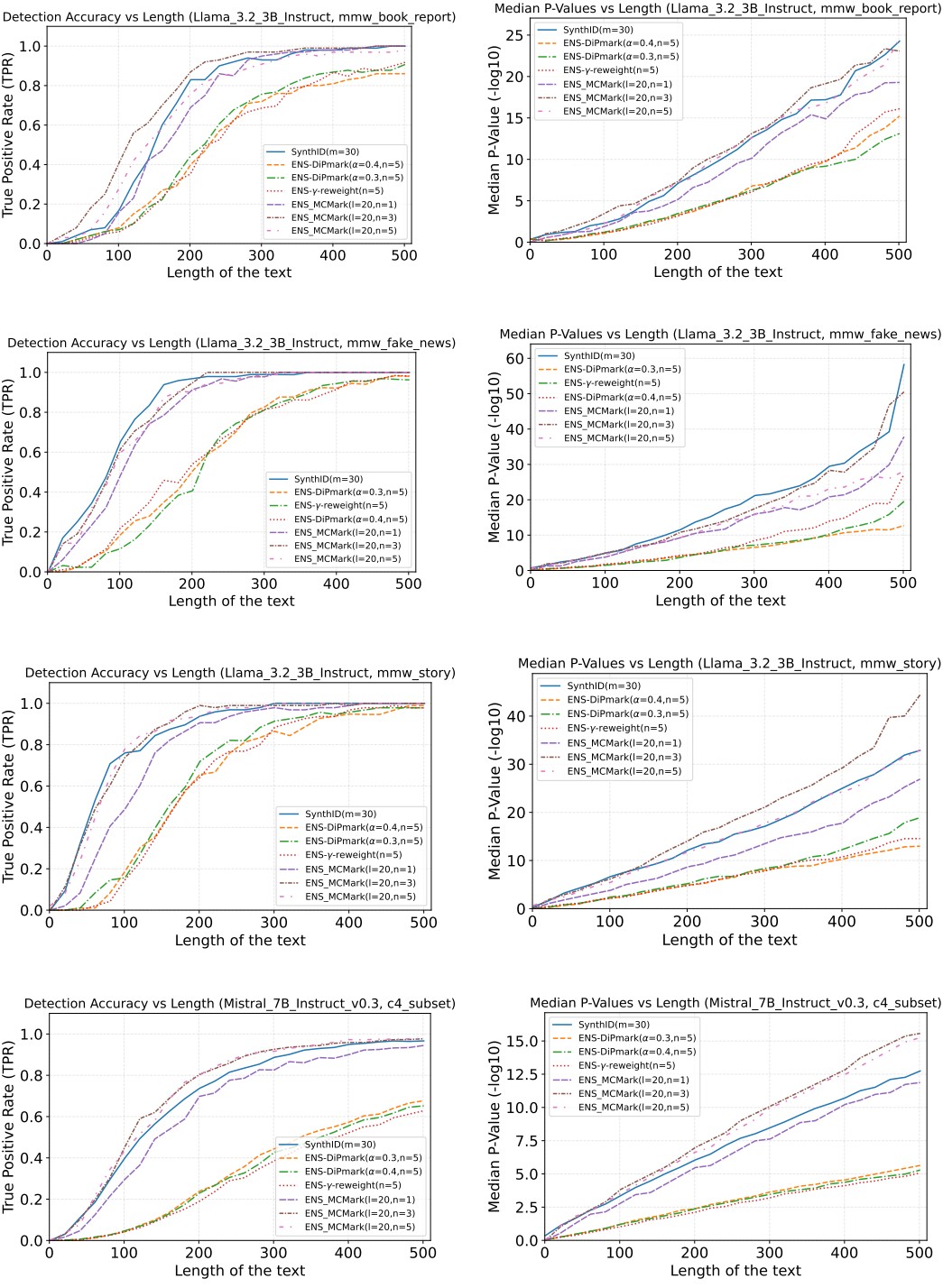

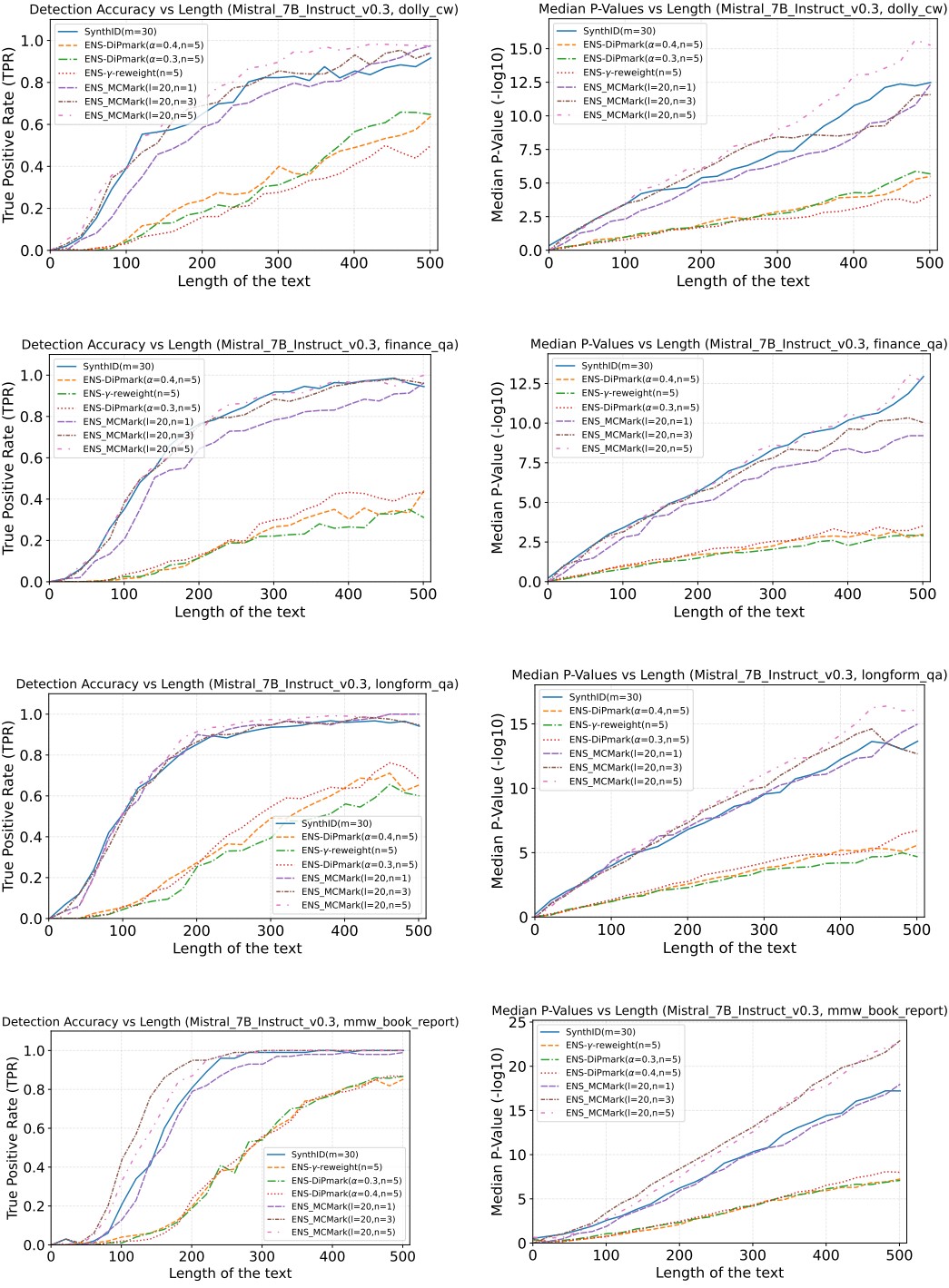

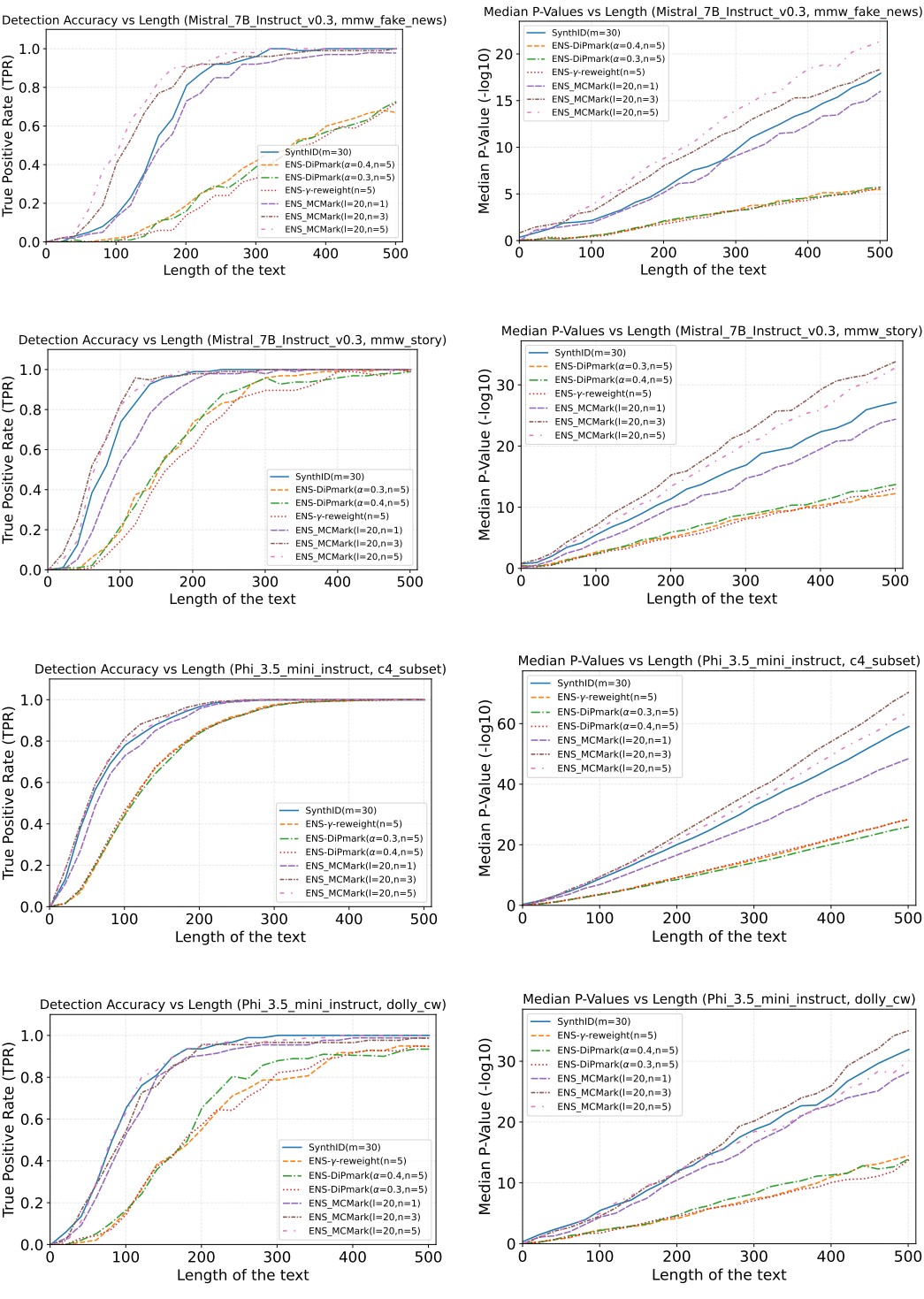

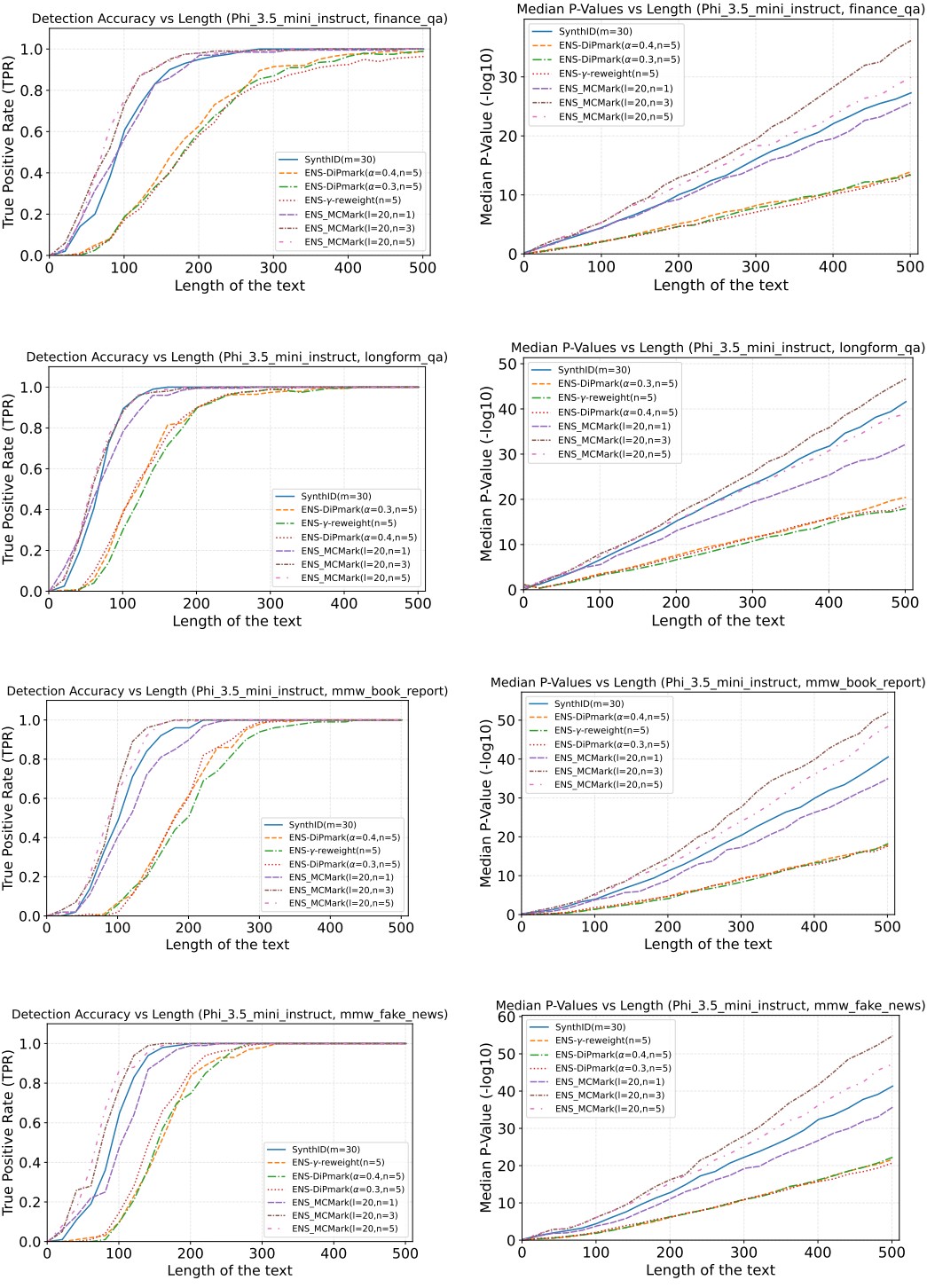

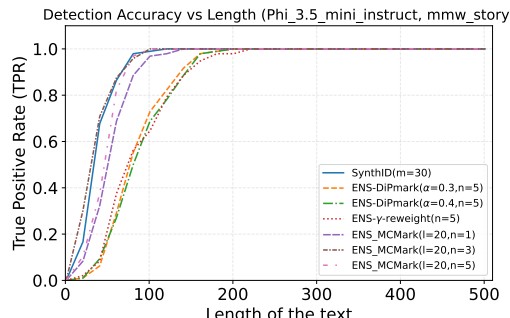 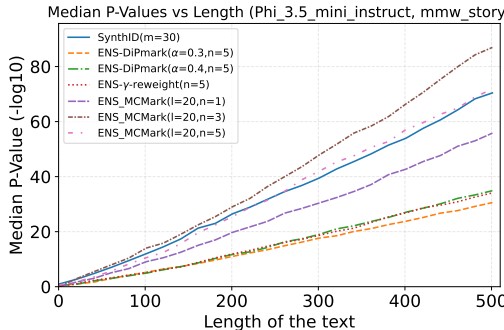

