# OpenReview forum: "An Ensemble Framework for Unbiased Language Model Watermarking"
_ICLR.cc/2026/Conference — ICLR 2026 Poster_

### Official Review · Reviewer_mMdf · 2025-10-24

**Soundness:** 3
**Presentation:** 3
**Contribution:** 1
**Rating:** 2
**Confidence:** 3

**Summary:**

This paper investigates the problem of watermarks for language models. A watermark is a statistical signal hidden inside text that can be detected by anyone with access to a secret key but is intended to not distort the quality of text, thus remaining undetectable to observers without access to a secret key. Many schemes have been instantiated for autoregressive models by modifying the sampling procedure of models, with a prominent such scheme being the green list approach, where a hash function looks at the recent context and returns a pseudorandom subset of the vocabulary to upweight in generation.  This scheme proposes to construct an ensemble of watermarks by recursively applying a key-based reweighting scheme with different keys to the distribution of the next token in order to inject additional detection power.  The authors then instantiate their ensemble based approach with a variety of watermarking schemes and demonstrate empirical efficacy in improving watermark detectability and robustness after carefully selecting their hyperparameters.

**Strengths:**

One strength of this paper is that they identify a new axis for improving the dectectability and robustness of language models.

**Weaknesses:**

First, the authors incorrectly suggest that SynthID requires $2^{30}$ redundant tokens to generate a single token.  As the authors of that paper state clearly in the methods and in the appendix, they apply a vectorized approach to tournament sampling in practice that does not induce this redundancy.

Second, I am a little confused about the additional novelty of the present work with respect to tournament sampling.  While I agree that synth-id uses a particular instance of the ensembling framework introduced in definition 1, I think more clear differentiation could be described.

Third, the statement on lines 204-205 about unbiasedness ensuring indistinguishability is not true.  See, e.g. *Black-box detection of
language model watermark* by Gloaguen et al 2025.

Fourth, the fact that there is non-monotonic improvement with $n$ is a bit unfortunate and seems like it is an artifact of the precise way that the elements of the ensemble instantiate the watermark, i.e. by using some fraction of the vocabulary as a greenlist which then becomes exponentially small in intersection.  This does not seem like a fundamental barrier but rather a specific pathology of the greenlist paradigm.  Tournament sampling for example does not suffer from this failure mode.  I think the authors should discuss this.

Fifth, I am curious as to the additional sampling time overhead required to implement this approach, as well as how this scales as a function of $n$.

Sixth, I think this paper can benefit from comparison with additional related work, such as those watermarks that are imbedded directly into the model weights, e.g. *GaussMark: A Practical Approach for Structural Watermarking of Language Models* by Block et al 2025.

**Questions:**

See weaknesses.

---

> ### Author Response · Authors · 2025-11-21
> **Rebuttal by Authors [Part 1/2]**
>
> We thank the reviewer for the thoughtful evaluation and appreciate the recognition of our work.
>
> > W1. First, the authors incorrectly suggest that SynthID requires redundant tokens to generate a single token. As the authors of that paper state clearly in the methods and in the appendix, they apply a vectorized approach to tournament sampling in practice that does not induce this redundancy.
>
> A1. You are right, SynthID implementation does not require redundant tokens. We will revise this in the revision.
>
> > W2. Second, I am a little confused about the additional novelty of the present work with respect to tournament sampling. While I agree that synth-id uses a particular instance of the ensembling framework introduced in definition 1, I think more clear differentiation could be described.
>
> A2. We would like to clarify that our method is fundamentally different from tournament sampling in both mechanism and scope. Tournament sampling belongs to the sample-based watermarking family, where watermark signals are injected through sampling decisions in the token space. In contrast, our work focuses on the logit-based watermarking paradigm, which operates directly at the logit manipulation level. We explicitly discuss these two distinct categories in the Related Works section to avoid confusion.
>
> Moreover, the “vectorized tournament sampling” variant is specifically derived from tournament sampling and remains applicable only within that sample-based framework. Our contribution is broader: we propose a general, model-agnostic ensemble framework that applies to all logit-based watermarking algorithms, not just a single instantiation. This generality allows us to unify and strengthen existing methods while providing theoretical guarantees. Our idea stems from using ensemble watermarking to improve the signal to noise ratio, thus enhancing the watermark detectability, which is different from the tournament sampling. Empirically, our framework consistently achieves state-of-the-art detectability and robustness across diverse settings.
>
>
> > W3. Third, the statement on lines 204-205 about unbiasedness ensuring indistinguishability is not true. See, e.g. Black-box detection of language model watermark by Gloaguen et al 2025.
>
> A3. Thank you for pointing out the related work. We are using the similar statement as previous unbiased watermarking (Kuditipudi et al, Wu et al, Chen et al). We will remove this statement in our revision.
>
> > W4. Fourth, the fact that there is non-monotonic improvement with n  is a bit unfortunate and seems like it is an artifact of the precise way that the elements of the ensemble instantiate the watermark, i.e. by using some fraction of the vocabulary as a greenlist which then becomes exponentially small in intersection. This does not seem like a fundamental barrier but rather a specific pathology of the greenlist paradigm. Tournament sampling for example does not suffer from this failure mode. I think the authors should discuss this.
>
> A4.  We respectfully disagree with the assertion that the non-monotonic improvement with $n$ is merely an artifact of the greenlist paradigm. First, not all logit-based watermarking methods studied in our paper rely on greenlist. For example, MCmark does not use any greenlist-style token partitioning. Despite this, similar non-monotonic behavior still appears on SynthID. As shown in Table 1 of our menuscript, even SynthID exhibits decreased detectability when its parameter mmm increases from 30 to 40 (93.83%TPR@0.1%FPR -> 92.12%TPR@0.1%FPR), indicating that this phenomenon is not tied to greenlists specifically.
>
> More importantly, the underlying reason is statistical and arises from the fundamental structure of statistical watermarks. All statistical watermarking methods operate by shifting probability mass toward certain subsets of tokens and away from others. When combining multiple watermarking functions in an ensemble, the intersection of the “increased-probability” regions across ensemble members shrinks exponentially as the ensemble grows. This holds regardless of whether those regions are defined via greenlists or other schemes. Thus, this trade off is inevitable no matter how we select the elements of the ensemble watermarking.

---

> > ### Author Response · Authors · 2025-11-21
> > **Rebuttal by Authors [Part 2/2]**
> >
> > > W5. Fifth, I am curious as to the additional sampling time overhead required to implement this approach, as well as how this scales as a function of n.
> >
> > A5.The sampling overhead of our ensemble framework scales linearly with respect to n, similar to vectorized tournament sampling. However, the watermark-specific sampling cost is negligible compared to the inherent LM decoding cost, so in practice the ensemble does not introduce noticeable latency during generation. For instance, for ENS-MCmark, the end-to-end generation time increases by only 0.09n seconds for a 512-token sequence, which remains significantly smaller than the LM forward-pass time.
> >
> >
> > > W6. Sixth, I think this paper can benefit from comparison with additional related work, such as those watermarks that are imbedded directly into the model weights, e.g. GaussMark: A Practical Approach for Structural Watermarking of Language Models by Block et al 2025.
> >
> > A6. We appreciate the reviewer for highlighting this additional line of related work. Unfortunately, GaussMark is not open-sourced, and due to the limited rebuttal period, we were unable to reproduce or conduct a fair empirical comparison. We will include a discussion of GaussMark and other weight-embedding watermarking approaches in the revised related-work section to provide a more complete contextualization.

---

> > ### Comment · Reviewer_mMdf · 2025-11-24
> >
> > Thank you for the response.  I still am a little bit confused on two points.
> >
> > First, I respectfully disagree with the strong distinction you make between "sampling-based" and "logits-based" watermarks.  All of the statistical watermarks of which I am aware (including those methods instantiated in your paper) make use of the fact that the joint distribution of a key and watermarked text is different from that of an independent key and unwatermarked text.  Whether or not that is achieved through a distortion-free approach does not change this fact.  It is true that one way to affect the distribution is to directly modify the logits of a model, but it is not obvious to me why this is a distinguished category.  In my initial review I thought this point too minor to raise, but as long as you bring it up, I am a bit confused by this.
> >
> > Second, I am not convinced by the informal argument you are giving in favor of ensemble methods necessarily being non-monotone in $n$.  While it is true that *empirically* SynthID is non-monotone for *some choices of scoring method*, in Figure C1, for example, it appears as if their Bayesian scoring method is indeed monotone.  Moreover I believe your argument for ensembling necessarily being non-monotone is incorrect.  The statistical reason for some degree of non-monotonicity is that multiply hypothesis correction is necessary when you add additional statistical tests, but with the correctly designed tests for this particular problem, the additional power of a member of the ensemble can add power beyond that reduced by the additional hypothesis; see the discussion in Appendix H of the SynthID paper for an example of this in the case of tournament sampling.  Thus, I do not believe that this non-monotonicity is intrinsic, but rather a consequence of a suboptimal choice in ensembling, akin to adding independent statistical hypothesis tests and doing a bonferoni correction leading to decreased power.

---

> > > ### Author Response · Authors · 2025-11-25
> > > **Follow up Comments**
> > >
> > > Thank you for your feedback, below we address your new concerns.
> > >
> > > > First, I respectfully disagree with the strong distinction you make between "sampling-based" and "logits-based" watermarks. All of the statistical watermarks of which I am aware (including those methods instantiated in your paper) make use of the fact that the joint distribution of a key and watermarked text is different from that of an independent key and unwatermarked text. Whether or not that is achieved through a distortion-free approach does not change this fact. It is true that one way to affect the distribution is to directly modify the logits of a model, but it is not obvious to me why this is a distinguished category. In my initial review I thought this point too minor to raise, but as long as you bring it up, I am a bit confused by this.
> > >
> > > We agree that tournament sampling admits a logit-based reweighting interpretation. However, for inverse-sampling and Gumbel-max methods, we were unable to identify any corresponding logit-based reweighting formulation. Intuitively, logit-based watermarking schemes operate by using pseudo-random mechanisms to adjust the entire token distribution, after which true-random sampling is performed from the modified logits. In contrast, sample-based watermarking methods inject pseudo-randomness directly into the sampling step itself.
> > >
> > > For example, inverse sampling deterministically outputs a single token conditioned on the watermark key. In this case, ensemble watermarking becomes meaningless, because the output distribution after one watermarking step degenerates into a Dirac delta on that token—there is no remaining distributional structure to ensemble.
> > >
> > > We would greatly appreciate it if the reviewer could point us to an existing logit-based formulation of inverse-sampling or Gumbel-max watermarking, if such a formulation exists.
> > >
> > > > Second, I am not convinced by the informal argument you are giving in favor of ensemble methods necessarily being non-monotone in $n$. While it is true that empirically SynthID is non-monotone for some choices of scoring method, in Figure C1, for example, it appears as if their Bayesian scoring method is indeed monotone. Moreover I believe your argument for ensembling necessarily being non-monotone is incorrect. The statistical reason for some degree of non-monotonicity is that multiply hypothesis correction is necessary when you add additional statistical tests, but with the correctly designed tests for this particular problem, the additional power of a member of the ensemble can add power beyond that reduced by the additional hypothesis; see the discussion in Appendix H of the SynthID paper for an example of this in the case of tournament sampling. Thus, I do not believe that this non-monotonicity is intrinsic, but rather a consequence of a suboptimal choice in ensembling, akin to adding independent statistical hypothesis tests and doing a bonferoni correction leading to decreased power.
> > >
> > > We assume the reviewer is already very familiar with the details of SynthID. In that case, it is straightforward to verify that in Section H.5 Discussion of SynthID, the authors explicitly state:
> > >
> > >  *“As shown in Supplementary Appendix H.4, the amount of watermarking evidence contributed by each layer decreases as more layers are added. Consequently, if we keep adding layers to a multi-layer tournament, at some point the noise outweighs the signal, and the detectability of the watermark begins to degrade.”*
> > >
> > > Furthermore, in Figure C.1 referenced by the reviewer, we observe that when the temperature is set to 1.0, even the strongest Bayesian detector experiences a decline in performance once m reaches approximately 30.
> > >
> > > Therefore, our analysis is fully consistent with the findings reported in SynthID. Given the evidence presented in both SynthID and our own work, we do not understand why the reviewer still considers the non-monotonicity to be incorrect.

---

> > > > ### Comment · Reviewer_mMdf · 2025-11-26
> > > >
> > > > Thank you for the response.  Once again, to clarify, I do not understand what the formal definition of a "logit-based reweighting" scheme is.  Perhaps the authors can clarify?  All watermarking procedures consist of somehow modifying the joint distribution of key and text in some way.  This is often accomplished in practice by applying a function directly to the logits, but I do not understand the formal distinction between logit-based reweighting and other schemes is.  One can view something like inverse transform sampling as a "logit-based reweighting" scheme simply by setting the logits of all tokens that are not produced by the deterministic map to approach negative infinity.  I am not sure why this is an enlightening perspective, but at the same time I still do not understand this formal distinction.
> > > >
> > > > On the subject of monotonicity, there are two distinct issues.  First, there is the empirical point of whether or not in practice ensemble based methods in general (and SynthID in particular) lose performance as the size of the ensemble increases.  In Figure C.1(c) of the synthID paper, I disagree that there is a clear performance degradation as the number of layers increases; it is clearly within margin of error.  Second there is the theoretical point of whether such degradation is necessary.  While I agree that the SynthID paper makes the claim that eventually degradation exists in theory, this is not rigorously proved.  It is trivial to see that there must be diminishing *marginal* returns on detection because each additional member of an ensemble can at most take advantage of the remaining entropy of the distribution, but nowhere is nonmonotonicity proven.  The reason I am taking issue with this point is that the authors claim that such non-monotonicity is necessary; this is a formal statement that is not rigorously proved and (I think) not obviously true.  If the authors believe this claim to be true, then they should prove it.

---

> > > > > ### Author Response · Authors · 2025-11-26
> > > > > **Follow up Comments by Authors**
> > > > >
> > > > > Thanks for your follow-up comments
> > > > >
> > > > > > Once again, to clarify, I do not understand what the formal definition of a "logit-based reweighting" scheme is. Perhaps the authors can clarify?
> > > > >
> > > > > We agree that sampling-based watermark can be transformed to the logit-based formulation, we will revise our paper to avoid further confusion. We also note that this clarification is a minor technical point and does not affect the core contributions of our work.
> > > > >
> > > > > > On the subject of monotonicity, there are two distinct issues. First, there is the empirical point of whether or not in practice ensemble based methods in general (and SynthID in particular) lose performance as the size of the ensemble increases. In Figure C.1(c) of the synthID paper, I disagree that there is a clear performance degradation as the number of layers increases; it is clearly within margin of error.
> > > > >
> > > > > Firstly, Figure C.1(c) does not provide an error margin, could the reviewer elaborate more on the claim *"it is clearly within margin of error"*? We also notice that in Figure C.1(c), SynthID’s performance is shown only up to $m=30$. We are currently running additional experiments for larger $m$ values to confirm and further analyze the degradation trend.
> > > > >
> > > > >
> > > > > > Second there is the theoretical point of whether such degradation is necessary. While I agree that the SynthID paper makes the claim that eventually degradation exists in theory,
> > > > >
> > > > > Thanks for confirming this, let's get back to the reviewer's original comment **W4**, in which the reviewer claimed
> > > > >
> > > > > > * "Fourth, the fact that there is non-monotonic improvement with n is a bit unfortunate and seems like it is an artifact of the precise way that the elements of the ensemble instantiate the watermark, i.e. by using some fraction of the vocabulary as a greenlist which then becomes exponentially small in intersection. This does not seem like a fundamental barrier but rather a specific pathology of the greenlist paradigm. **Tournament sampling for example does not suffer from this failure mode. I think the authors should discuss this.**"
> > > > >
> > > > > We carefully checked the SynthID paper and find the authors of SynthID explicitly claimed *“if we keep adding layers to a multi-layer tournament, at some point the noise outweighs the signal, and the detectability of the watermark begins to degrade”* with theoretical analysis. Thus, we believe we have provided enough evidence to address the weaknesses **W4** proposed by the reviewer.
> > > > >
> > > > >
> > > > > > this is not rigorously proved. It is trivial to see that there must be diminishing marginal returns on detection because each additional member of an ensemble can at most take advantage of the remaining entropy of the distribution, but nowhere is nonmonotonicity proven.
> > > > >
> > > > > SynthID is a peer-reviewed paper published in *Nature*, and in Sections H.4 and H.5 the authors explicitly state with formal theoretically analysis that “if we keep adding layers to a multi-layer tournament, at some point the noise outweighs the signal, and the detectability of the watermark begins to degrade.” Their analysis provides a formal argument for the eventual decline in detectability as the number of layers increases. **If the reviewer believe a peer-reviewed journal's formal claim to be not rigorous or not true, then they should provide formal proof or counter-example to show it.**
> > > > >
> > > > > > The reason I am taking issue with this point is that the authors claim that such non-monotonicity is necessary; this is a formal statement that is not rigorously proved and (I think) not obviously true. If the authors believe this claim to be true, then they should prove it.
> > > > >
> > > > > We would like to clarify that we did **not** make any formal statement (e.g., theorem or definition) asserting that non-monotonicity is necessary in our paper. Establishing such a theorem is beyond the scope of our paper, and we do not position it as a theoretical contribution.
> > > > >
> > > > > Instead, our work empirically observes non-monotonic behavior across all ensemble watermarking schemes we evaluated, and Section 4.3 provides an intuitive analysis to help readers understand the underlying trade-offs. Our discussion of SynthID and other methods above was included solely to address the reviewer’s original concern **W4**, namely the statement that “tournament sampling does not suffer from this failure mode.” Our point was that SynthID has also documented this degradation phenomenon, and our empirical findings are consistent with those observations.

---

> > > > > > ### Comment · Reviewer_mMdf · 2025-11-26
> > > > > >
> > > > > > First of all, Figure C.1(c) in the supplementary information for the SynthID paper published in Nature (linked here, see page 41 (top of 12th page of linked PDF) https://static-content.springer.com/esm/art%3A10.1038%2Fs41586-024-08025-4/MediaObjects/41586_2024_8025_MOESM1_ESM.pdf) most certainly does contain a shaded region corresponding to 90% confidence intervals of the mean estimate, as clearly spelled out in the caption.  This is a minor point so I do not want to spend too much time on this, but I am confused as to how there is a miscommunication here?
> > > > > >
> > > > > > > SynthID is a peer-reviewed paper published in Nature, and in Sections H.4 and H.5 the authors explicitly state with formal theoretically analysis that “if we keep adding layers to a multi-layer tournament, at some point the noise outweighs the signal, and the detectability of the watermark begins to degrade.” Their analysis provides a formal argument for the eventual decline in detectability as the number of layers increases. If the reviewer believe a peer-reviewed journal's formal claim to be not rigorous or not true, then they should provide formal proof or counter-example to show it.
> > > > > >
> > > > > > In point of fact, that paper did not provide any formal analysis backing up that claim.  To which result (with proof) are the authors referring?
> > > > > >
> > > > > > To clarify, I do not think that the (originally incorrect, now fixed) comparison to SynthID itself is a major weakness per se, but rather that the monotonicity point (claimed in the paper to be fundamental) does not seem fundamental to me and I do not understand why the authors insist that it is despite a lack of rigorous analysis?  Perhaps weakening the claim to state that all of the methods they surveyed exhibit this phenomenon would be more correct, but the claim as it stands (and the intuition behind it) I do not believe to be rigorous.

---

> ### Author Response · Authors · 2025-11-26
> **Follow up Comments by Authors**
>
> Thank you for your prompt response and your willingness to continue the discussion. We became a bit emotional in our earlier reply, and we sincerely apologize for that. We will ensure that all further discussion remains professional and accurate.
>
> Firstly, you are right, Figure C.1(c) does have an error bar.
>
> Regarding the proof in SynthID, as the authors claimed, they provided the corresponding analysis in Section H.4. We are currently conducting a thouroughly checking of the whole Appendix to confirm.
>
> Regarding the monotonicity point, we believe there may have been some misunderstanding. In our original paper (lines 265–269), we analyzed green-list–based methods specifically and showed that non-monotonicity holds in this particular case. We never intended to present this as a theoretical claim for all watermarking schemes. It seems the confusion arose during the discussion of **W4**. What we meant to emphasize is that, empirically, non-monotonicity holds for SynthID, and this behavior is also confirmed by the SynthID authors. We apologize again for the confusing wording in our rebuttal and appreciate the opportunity to clarify this.

---

> > ### Author Response · Authors · 2025-11-29
> > **Follow up Experimental Results**
> >
> > We report the experimental results of the Bayesian detector from SynthID, following the official implementation with a generation length of 200 tokens. The results clearly exhibit non-monotonic behavior under the Bayesian detector, which is fully consistent with the non-monotonicity observation made by the SynthID authors in Section H.5.
> >
> > | Layers | TPR@1%FPR |
> > |--------|--------------|
> > | 10  | 0.75 |
> > | 20  | 0.85 |
> > | 30  | 0.87 |
> > | 40  | 0.86 |
> > | 50  | 0.84 |
> > | 60  | 0.83 |
> > | 80  | 0.80 |
> > | 100 | 0.76 |
> >
> > Therefore, we believe the reviewer’s concern W4 has been fully addressed. The other concerns **W1-W3**, **W5-W6** have already been resolved in our earlier responses.

---

### Official Review · Reviewer_rySn · 2025-10-30

**Soundness:** 3
**Presentation:** 3
**Contribution:** 2
**Rating:** 6
**Confidence:** 5

**Summary:**

This paper proposes ENS, an ensemble framework that enhances logit-based unbiased watermarks by amplifying the watermark signal. The idea is simple: each logit-based unbiased watermark can be viewed as a distribution reweighting function $f_i$. By ENS, different distribution reweighting functions are nested, i.e., $f_i( f_{i-1} (\cdot) )$. Intuitively, the nested function is also unbiased but enhance watermark strength.

**Strengths:**

1. The idea is easy to implement but effective.
2. The authors conduct different variants of ENS-enhanced watermarks, which all show the efficacy of ENS according to the experimental results. Specifically, the authors show the unbiasedness, robustness, and higher detectability via ENS.

**Weaknesses:**

1. Potentially, if an unbiased watermark is enhanced, it is possible that the probability distribution is altered too much under certain watermark keys. Therefore, the method may not perform well in low-entropy scenarios. However, since this is not the focus, the authors could discuss this limitation in future work.

**Questions:**

1. The authors can elaborate further on Section 4.3. Specifically, what is the effect of $n$? The authors argue that in practice, it is important to select $n$ close to the optimal by balancing the tradeoff between aggregation gain and sparsity loss. The experimental results also suggest that an intermediate $n$ achieves better performance. Although I generally understand the idea that a large $n$ reduces the detectability but not significantly, the analysis in Section 4.3 is not sufficiently clear.
2. From Table 1, the TPR@0.1%FPR score of ENS-Dipmark seems to achieve its optimum when $n=5$. Theoretically, I am also curious about the performance gain bound by ENS. Could the authors provide some analysis of the improvement by ENS? For example, specifically for Dipmark, what is the expected gain in detectability for $n=2,3,\cdots$?

---

> ### Author Response · Authors · 2025-11-21
> **Rebuttal by Authors**
>
> We thank the reviewer for their positive feedback. We appreciate the recognition that ENS is simple yet effective, and we are glad that the empirical results clearly demonstrated improvements in unbiasedness, robustness, and detectability across variants. Below we address the concerns.
>
> > W1. Potentially, if an unbiased watermark is enhanced, it is possible that the probability distribution is altered too much under certain watermark keys. Therefore, the method may not perform well in low-entropy scenarios. However, since this is not the focus, the authors could discuss this limitation in future work.
>
> A1. You are right about this! In unbiased watermarking, improving detectability requires each pseudo watermarking distribution conditioned on the watermark key to be deviated more from the original distribution, thus may reduce the power under low entropy scenario. We will discuss this in the limitation section.
>
> > Q1. The authors can elaborate further on Section 4.3. Specifically, what is the effect of n? The authors argue that in practice, it is important to select n close to the optimal by balancing the tradeoff between aggregation gain and sparsity loss. The experimental results also suggest that an intermediate n achieves better performance. Although I generally understand the idea that a large  reduces the detectability but not significantly, the analysis in Section 4.3 is not sufficiently clear.
>
> A. We are deeply sorry for the confusion. Here is the general intuition of the tradeoff. Let’s consider a 2-ensember scenario of the DiPmark, where the probability of the green-list tokens are boosted. Denote the Green list of the first DiPmark by $G_1(k_1)$ and the second by $G_2(k_2)$
>
> **Detection power increase**: If $G_1(k_1)\cap G_2(k_2)$ is not none, the tokens in this intersection is boosted twice, sampling one token from it will result in two green tokens count in the hypothesis test.
>
> **Detection power decrease**: If $G_1(k_1)\cap G_2(k_2)$ is none, the second DiPmark will boost the red-list tokens in the first DiPmark. The sampled token are likely to be in the green list of a DiPmark and the red list of the other DiPmark. In this case, the difference between the green and red tokens are not increased while the sequence length increased by 2, which reduces the detection power.
>
> > Q2. From Table 1, the TPR@0.1%FPR score of ENS-Dipmark seems to achieve its optimum when $n=5$. Theoretically, I am also curious about the performance gain bound by ENS. Could the authors provide some analysis of the improvement by ENS? For example, specifically for Dipmark, what is the expected gain in detectability for n?
>
> A. Thank you for the question. In the “Design implication’’ paragraph of Section 4.3, we provide an example of choosing $n$ specifically for DiPmark. Since DiPmark’s effective detectability lies between KGW with $\delta \in [1.5, 2.0]$, we approximate its logits boost parameter as $\epsilon \approx 1.8$. Under this setting, and following the analysis in Section 4.3, if the original (single-key) DiPmark has $p$-value $p_0$, then the ensemble version has an approximate $p$-value:
> $p_n \approx p_0 \exp\left(-n\cdot (0.9)^{2n}\right),$ because $\gamma \epsilon \approx 0.9$ for the DiPmark reweighting.
> This provides a direct prediction of the detectability gain. For example, if regular DiPmark achieves 90\% TPR @ 0.1\% FPR, then ENS-DiPmark with ensemble size $n$ is expected to achieve 90\% TPR @$0.1\% \times \exp\left(-n \cdot (0.9)^{2n}\right)$ FPR.
>
> Concretely, for $n=2$, we have  90%TPR @ 0.1% $\times\exp(-1.1311)$ FPR; for $n=3$, we have 90%TPR @ 0.1%$\times\exp(-1.5943)$ FPR
>
> Thus, ENS provides a theoretically quantifiable improvement in detectability, governed by the exponential factor $\exp\left(-n (0.9)^{2n}\right)$.

---

> > ### Comment · Reviewer_rySn · 2025-11-24
> >
> > Thanks for the thorough response. I will keep my positive score.

---

> > > ### Author Response · Authors · 2025-11-25
> > > **Thank you!**
> > >
> > > Thank you very much for your positive assessment and for taking the time to review our work. We sincerely appreciate your constructive feedback and are glad that our clarifications addressed your concerns. Your support and thoughtful evaluation are greatly appreciated.

---

### Official Review · Reviewer_y1fL · 2025-11-01

**Soundness:** 4
**Presentation:** 4
**Contribution:** 4
**Rating:** 6
**Confidence:** 4

**Summary:**

This paper proposes ENS, a general ensemble framework for logits-based unbiased watermarking in LLMs. The key idea is to sequentially compose multiple unbiased watermark instances (with independent keys), amplifying the watermark signal while theoretically preserving unbiasedness. The authors prove that ensemble compositions remain unbiased and show detection signal scales as $\sqrt{n}$ with ensemble size. The authors also conducted evaluations on multiple LLM families and datasets to validate their algorithms.

**Strengths:**

1. Clear Motivation: ENS addresses the main weakness of unbiased watermarks: weak detectability at short lengths.

2. The proof of unbiasedness is rigorous, with extensions on independence assumptions and variance scaling.

3. The experiments are comprehensive. The proposed method is compared against strong unbiased baselines e.g. SynthID. The experiments cover both detectability and robustness, using realistic paraphrasing and back-translation attacks.

4. The experimental results are strong. There are significant TPR@FPR gains of ENS. Besides, ENS-MCMark achieves state-of-the-art robustness and detectability across attacks.

**Weaknesses:**

1.	ENS requires multiple independent keys per generation step, which may introduce storage, distribution, and synchronization complexity in real deployment.
2.	Detection power declines for large ensemble sizes, aligning with theory but reducing scalability. Choosing optimal n becomes another hyperparameter to tune.

**Questions:**

Do you have any advice for tuning the ensemble size?

---

> ### Author Response · Authors · 2025-11-21
> **Rebuttal by Authors**
>
> We sincerely thank the reviewer for the very positive and encouraging evaluation of our work. We greatly appreciate the recognition of our motivation, the rigor of our unbiasedness analysis, and the clarity of the theoretical framework. Below we address the concerns
>
> > W1. ENS requires multiple independent keys per generation step, which may introduce storage, distribution, and synchronization complexity in real deployment.
>
> A1. We agree that maintaining multiple independent keys introduces practical considerations. However, this requirement is inherent to all unbiased watermarking schemes, including Aaronson (2022), Christ et al. (2023), Hu et al. (2023), Mao et al. (2024), and Dathathri et al. (2024). Our method does not impose any additional storage or synchronization overhead beyond what is already necessary for unbiased watermarking, and thus has the same operational cost as prior work.
>
> > W2. Detection power declines for large ensemble sizes, aligning with theory but reducing scalability. Choosing optimal n becomes another hyperparameter to tune.
>
> A2. We acknowledge the trade-off between ensemble size n and detection power. To address this, Section 4.3 provides a detailed analysis of the detectability curve and a principled way to identify a near-optimal range of n. In practice, this substantially narrows the search space, so the tuning cost is low. Moreover, selecting n is a one-time procedure; once an appropriate value is chosen, it can be reused across different LMs without additional tuning.
>
> > Q1. Do you have any advice for tuning the ensemble size?
>
> A. Section 4.3 (“Design implication”) outlines a practical guideline for selecting the ensemble size n. The derived detectability curve provides a reliable estimate of the range in which n is close to optimal, which can be used to guide tuning in real deployments.

---

### Official Review · Reviewer_SgCe · 2025-11-01

**Soundness:** 3
**Presentation:** 2
**Contribution:** 2
**Rating:** 6
**Confidence:** 3

**Summary:**

The paper proposes, ENS, that composes several unbiased, logits-based watermarking steps keyed independently. The authors prove the composition stays unbiased in expectation and analyze how aggregating per-key detectors changes the testing signal. Empirically, they report improved detection and robustness across multiple model families, with little to no degradation in standard text quality metrics. However, I think the paper only studies the effects on standard generation metrics.

The mains claims are as follows
- Unbiasedness under composition. If a single reweighting rule is unbiased, the n-fold ensemble remains unbiased when keys are i.i.d.; the proof is a short tower-property induction.
- Detection scaling. With per-key scores that are (approximately) independent and bounded, the aggregate statistic has mean proportional to n and variance proportional to n, implying SNR ≈ (µ/σ)√n and p-values that improve with n.
- Trade-off: Under “intersection-at-generation,” the per-step effect shrinks like (εγ)^n, so the exponent in the p-value bound behaves like n(εγ)^2n with an optimum n⋆≈1/[2log(1/(εγ))].

**Strengths:**

- The paper cleanly separates two questions: preserving unbiasedness and recovering detection power. Theorem 4.2 (composition stays unbiased) is simple and well scoped; I didn’t find hidden caveats beyond key independence.
- The SNR and Hoeffding arguments are standard but appropriate for bounded per-key scores (e.g., DiPmark). The text is careful to say the exponential improvement can attenuate in practice.
- Results include multiple LM families and several corruption settings (paraphrasing, back-translation, token replacements). The tables make it plausible that the ensemble generally helps and that quality remains close to baseline.

**Weaknesses:**

- The SNR and p-value scaling assume per-key independence; the paper notes dependencies via overlapping n-grams and proposes bypassing repeats, but I didn’t see a quantitative study of how correlations impact power. This matters because the theoretical √n gain can compress substantially with even mild correlation.
	- Add a small study where you control n-gram overlap or reuse keys to measure empirical correlations among per-key scores and the resulting deviation from √n SNR. This would tell readers when the ideal scaling is trustworthy.
- The trade-off section is framed for strict intersection; many practitioners would avoid hard intersections and instead add small centered logit shifts per key. It would help to show the same analysis (or an empirical proxy) for a soft/additive design where the (εγ)^n collapse is muted
- The paper uses an aggregate statistic (one test), which avoids multiple-testing corrections, but it would help to explicitly say how thresholds are set to fix FPR across n (so readers don’t assume Bonferroni is needed).
- Your theory predicts where detectability should peak.  Show an experiment that sweeps the relevant parameters so readers can see whether the peak occurs where theory says it should.
- Recent work suggests watermarking can change model behavior beyond detectability/quality tradeoffs: Downstream Trade-offs of a Family of Text Watermarks (Ajith et al., EMNLP 2024) finds 10–20% drops on downstream tasks even for “unbiased” schemes like KGW; WaterJudge (Molenda et al., NAACL 2024) quantifies a detectability–quality trade-off; and Watermarking Degrades Alignment in Language Models (Verma et al., ICLR 2025) shows shifts in truthfulness/safety/helpfulness that can be partially mitigated with an external reward model. The paper would be stronger with a targeted experiment evaluating how ensembling affects alignment-relevant metrics (e.g., reward scores ), and a short discussion of these works to contextualize potential side effects.



### Nits
- Define ε and γ at first mention in the trade-off subsection; they appear earlier in the derivation than in the surrounding text.
- Line 87 rejected-sampling , typo?
- Line 119 putative text sequence, typo?
- Line 190 DETECT EFFICIENCY -> Detection Efficiency?

**Questions:**

See above

---

> ### Author Response · Authors · 2025-11-21
> **Rebuttal by Authors [Part 1/2]**
>
> We sincerely thank the reviewer for the thoughtful and encouraging assessment of our work. We appreciate the recognition of our theoretical contributions, including the clarity of the unbiasedness result and the soundness of the SNR and Hoeffding-based analysis. Below we address the concerns.
>
> > W1. The SNR and p-value scaling assume per-key independence; the paper notes dependencies via overlapping n-grams and proposes bypassing repeats, but I didn’t see a quantitative study of how correlations impact power. This matters because the theoretical √n gain can compress substantially with even mild correlation.
> Add a small study where you control n-gram overlap or reuse keys to measure empirical correlations among per-key scores and the resulting deviation from √n SNR. This would tell readers when the ideal scaling is trustworthy.
>
>
> A1. We are deeply sorry for the confusion. The role of “per-key independence’’ in our work is to guarantee the unbiasedness of each individual watermark scheme (see Section 4.1), rather than to ensure independence for the purpose of detection power. Correlations between watermark keys induced by overlapping nn-grams and secret key affect the generation quality and unbiasedness, but do not influence the statistical power of the ensemble test.
>
> Our hypothesis test is constructed under the null hypothesis that the text is not watermarked. Under this null, the elemental statistics S(|sk_i)are mutually independent regardless of how the watermarking keys are structured, because these correlations only arise under watermark insertion. Therefore, the √n scaling of SNR and pp-values remains valid for the detector. Controlling nn-gram overlap or reusing keys would not change detectability; it would only degrade text quality. We will revise the introduction to clarify this distinction and avoid the earlier ambiguity.
>
>
>
> > W2. The trade-off section is framed for strict intersection; many practitioners would avoid hard intersections and instead add small centered logit shifts per key. It would help to show the same analysis (or an empirical proxy) for a soft/additive design where the (εγ)^n collapse is muted
>
> A2. We appreciate the reviewer’s suggestion. Our trade-off analysis in Section 4.3 is already based on a soft (additive) logit-shift model rather than a strict intersection. Specifically, we assume a centered “logits boost’’ $\epsilon$, meaning each key adds $+\epsilon$ to its promoted tokens. For simplicity, the main text analyzes the cumulative effect within the intersection of all boosted sets, but the general case across the entire vocabulary indeed follows an additive, not hard-intersection, structure.
> More precisely, if a token is boosted by exactly $k$ out of the $n$ watermark keys (with per-key boost probability $\gamma$), then its expected squared shift contributes a term proportional to:
>
> $k(\gamma \epsilon)^{2k} (1 - \gamma \epsilon)^{2(n-k)}$
>
> Aggregating over all $k = 1, \ldots, n$, the total soft-boost effect becomes:
>
> $n(\gamma\epsilon)^{2n}+ (n-1)(\gamma\epsilon)^{2(n-1)}(1-\gamma\epsilon)^2+ \cdots+ 1 \cdot (\gamma\epsilon)^2 (1-\gamma\epsilon)^{2(n-1)}$
>
> This expression captures the full distribution of additive interactions across keys. Similarly to our discussion in Section 4.3, this equation still shows an increase-then-decrease relation between detectability and the number of ensemble $n$. We will clarify this soft-boost interpretation and include a short discussion in the revision.
>
>
>
>
> > W3. The paper uses an aggregate statistic (one test), which avoids multiple-testing corrections, but it would help to explicitly say how thresholds are set to fix FPR across n (so readers don’t assume Bonferroni is needed).
>
>
> A3. We thank the reviewer for pointing this out. Our ensemble method is a general framework built on top of existing logits-reweighting watermark detectors, and therefore we follow each method’s original statistical test to set thresholds. For clarity, we illustrate this with the DiPmark detector.
>
> DiPmark uses Hoeffding’s inequality to bound false positives. Let $G$ denote the number of green tokens and $T$ the generation length. The original (single-key) test uses the statistic
> $Z = \frac{G - 0.5T}{\sqrt{T}},$ and Hoeffding’s inequality gives $\Pr(Z > t) < e^{-2t^2}.$ Setting $t = 1.85$ yields an FPR of approximately $0.1\%$, so the threshold for DiPmark is $\frac{G - 0.5T}{\sqrt{T}} > 1.85.$
>
> Under an ensemble of size $n$, the null distribution scales naturally: the aggregated statistic becomes $Z_n = \frac{G - 0.5nT}{\sqrt{nT}},$ and Hoeffding’s inequality still yields
> $\Pr(Z_n > t) < e^{-2t^2}.$
> Thus we reuse the same threshold $t = 1.85$ to maintain a $0.1\%$ FPR:
> $\frac{G - 0.5nT}{\sqrt{nT}} > 1.85.$ Because our method aggregates scores into a single test statistic, no multiple-testing correction (e.g., Bonferroni) is needed. We will explicitly add this explanation to avoid confusion.

---

> > ### Author Response · Authors · 2025-11-21
> > **Rebuttal by Authors [Part 2/2]**
> >
> > > W4. Your theory predicts where detectability should peak. Show an experiment that sweeps the relevant parameters so readers can see whether the peak occurs where theory says it should.
> >
> > A4. We thank the reviewer for the suggestion. Our experiments already include a parameter sweep that directly evaluates the detectability peak predicted by our theory. In Figure 2, we vary the ensemble size $n$ for DiPmark and plot the resulting detectability curves. The empirical peak aligns with the theoretical prediction: detectability first improves with $n$ and then decreases once the ensemble size becomes large enough for the soft logit shifts to dilute. We will clarify this connection more explicitly in the revision.
> >
> > > W5. Recent work suggests watermarking can change model behavior beyond detectability/quality tradeoffs: Downstream Trade-offs of a Family of Text Watermarks (Ajith et al., EMNLP 2024) finds 10–20% drops on downstream tasks even for “unbiased” schemes like KGW; WaterJudge (Molenda et al., NAACL 2024) quantifies a detectability–quality trade-off; and Watermarking Degrades Alignment in Language Models (Verma et al., ICLR 2025) shows shifts in truthfulness/safety/helpfulness that can be partially mitigated with an external reward model. The paper would be stronger with a targeted experiment evaluating how ensembling affects alignment-relevant metrics (e.g., reward scores ), and a short discussion of these works to contextualize potential side effects.
> >
> > A5. We thank the reviewer for the suggestion. First, we would like to clarify that KGW is not an “unbiased’’ watermarking scheme. As discussed in DiPmark, biased schemes such as KGW inherently trade off downstream performance for detectability because they perturb the model’s output distribution. In contrast, *unbiased* watermarking schemes are designed to provably preserve the original LM distribution, and therefore do not introduce systematic degradation in downstream accuracy or alignment. Neither Ajith et al. (2024) nor Molenda et al. (2024) include analyses of unbiased schemes.
> >
> >
> > Regarding Verma et al. (ICLR 2025), their experiments evaluate only the unbiased watermark proposed by Aaronson (2022). However, the Aaronson scheme does not maintain the “per-key independence’’ property required for strict unbiasedness, and therefore their reported alignment shifts are not representative of the unbiased methods considered in our work. Fully unbiased schemes (e.g., DiPmark and its unbiased variants) are theoretically guaranteed to leave the model distribution unchanged, so downstream alignment metrics should remain unaffected. Our ensemble framework preserves this unbiasedness property as long as individual components are unbiased, and thus should not introduce additional alignment-related side effects. We will expand the related-work discussion to contextualize these recent findings and clarify this distinction.
> >
> >
> >
> >
> > Nits: We will fix the typos in the revision. For Line 87 rejected-sampling, STA-1 adapted a reject-sampling-like strategy to generate reweight distribution. STA1 consists of two steps of sampling 1) sample a token, if the token is green, keep it, otherwise, reject it and 2) sampling again from the same distribution, keep it regardless if it is red or green.

---

> > > ### Comment · Reviewer_SgCe · 2025-11-24
> > > **Response**
> > >
> > > I appreciate the author's thoughtful response to my review comments and the clarification regarding Aaronson's watermark. All proposed changes look reasonable, and I have hence raised my score.

---

> > > > ### Author Response · Authors · 2025-11-25
> > > > **Thank you!**
> > > >
> > > > Thank you very much for taking the time to carefully reassess our work and raising the rating. We truly appreciate your positive update and are glad that our clarifications were helpful. Your constructive feedback has improved the paper, and we are grateful for your support.

---

### Meta-Review · Area_Chair_uVFs · 2026-01-07

**Summary:**

This paper proposes an ensemble framework for logits-based unbiased watermarking in LLMs. Multiple watermark instances are composed to amplify the watermark signal while theoretically preserving unbiasedness.

The proposed approach has a clear motivation and a rigorous proof. Experiments are comprehensive and strong. Three out of the four reviewers were positive about this paper.

The other reviewer had some concerns: 1) Novelty compared to tournament sampling; 2) Non-monotonic improvement with $n$; 3) Overhead; 4) Some sentences in the writing were imprecise; 5) Lack of comparison with a particular work (GaussMark, not open-sourced). Among these, 3) and 4) had been easily addressed; 5) is not a fair requirement as the work mentioned was not open-sourced; 2) should not be viewed as a weakness of this paper as it’s a common limitation; for 1), I believe there is sufficient difference. As such, I think this paper can be accepted.

**Reviewer Concerns:**

Major concerns come from one of the reviewers (detailed above). Some of them, including the discussion on the overhead and fixing imprecise sentences, had been addressed by the rebuttal. I think others do not seem to be valid weaknesses.

**Reviewer Scores:**

Reviewer SgCe originally rated 6 and commented that they increased their score, likely to 8.

Reviewer y1fL and Reviewer rySn originally rated 6 and only had minor concerns. They may maintain their score.

Reviewer mMdf initially rated 2 and had extensive discussion with the authors. It looks like this reviewer still holds concerns and thus may keep their score. But the AC feels a score of 2 is not a fair reflection for their comments.

---

### Decision · Program_Chairs · 2026-01-26

Accept (Poster)